# ADDRESSING THE TOPOLOGICAL DEFECTS OF DISENTANGLEMENT

## ABSTRACT

A core challenge in Machine Learning is to disentangle natural factors of variation in data (e.g. object shape *vs* pose). A popular approach to disentanglement consists in learning to map each of these factors to *distinct subspaces* of a model's latent representation. However, this approach has shown limited empirical success to date. Here, we show that this approach to disentanglement introduces *topological defects* (i.e. discontinuities in the encoder) for a broad family of transformations acting on images —encompassing simple affine transformations such as rotations and translations. Moreover, motivated by classical results from group representation theory, we propose an alternative, more flexible approach to disentanglement which relies on *distributed equivariant operators*, potentially acting on the *entire latent space*. We theoretically and empirically demonstrate the effectiveness of our approach to disentangle affine transformations. Our work lays a theoretical foundation for the recent success of a new generation of models using distributed operators for disentanglement (see discussion). All code is available at `https://anonymous.4open.science/r/5b7e2cbb-54dc-4fde-bc2c-8f75d29fc15a/`.

## 1 INTRODUCTION

Learning disentangled representations is arguably key to build robust, fair, and interpretable ML systems (Bengio et al., 2013; Lake et al., 2017; Locatello et al., 2019a). However, it remains unclear how to achieve disentanglement in practice. Current approaches aim to map different factors of variations in the data to distinct subspaces of a latent representation, but have achieved only limited empirical success (Higgins et al., 2016; Burgess et al., 2018). More work on the theoretical foundations of disentanglement could provide the key to the development of more successful approaches.

In its original formulation, disentanglement consists in isolating *statistically independent* factors of variation in data into independent latent dimensions. This perspective has led to a range of theoretical studies investigating the conditions under which these factors are identifiable (Locatello et al., 2019b; Shu et al., 2020; Locatello et al., 2020; Hauberg, 2019; Khemakhem et al., 2020). More recently, Higgins et al. (2018) has proposed an alternative perspective connecting disentanglement to group theory (see Appendix A for a primer on group theory). In this framework, the factors of variation are different *subgroups* acting on the dataset, and the goal is to learn representations where separated subspaces are *equivariant* to distinct subgroups —a promising formalism since many transformations found in the physical world are captured by group structures (Noether, 1915). However, the fundamental principles for how to design models capable of learning such equivariances remain to be discovered (but see Caselles-Dupré et al. (2019)).

Here we attack the problem of disentanglement through the lens of topology (Munkres, 2014). We show that for a very broad class of transformations acting on images —encompassing all affine transformations (e.g. translations, rotations), an encoder that would map these transformations into dedicated latent subspaces would necessarily be discontinuous. With this assurance, we reframe disentanglement by distinguishing its objective from its traditional implementation, resolving the discontinuities of the encoder. Guided by classical results from group representation theory

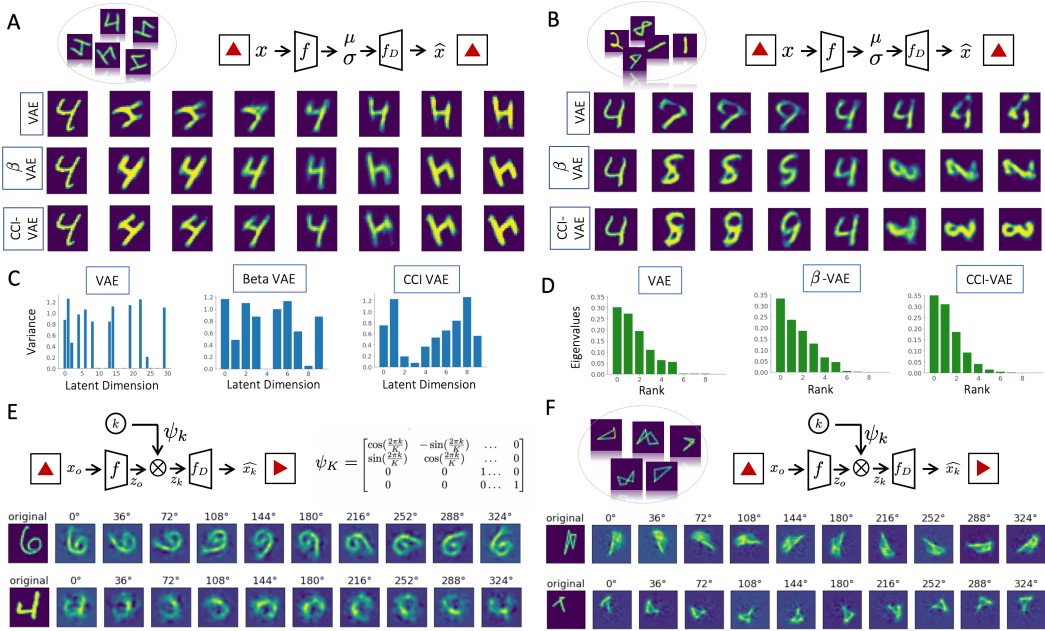

Figure 1: **Failure modes of common disentanglement approaches. A.** Latent traversal best capturing rotation for a VAE, $\beta$-VAE, and CCI-VAE for rotated MNIST restricted to a single digit class ("4"). **B.** Same as panel A for all 10 MNIST classes. **C.** Variance of single latents in response to image rotation, averaged over many test images. **D.** Ranked eigenvalues of the latent covariance matrix in response to image rotation, averaged over many test images. **E.** A supervised disentangling model successfully reconstructs some digits (top) but fails on other examples (bottom). **F.** Failure cases of the supervised model trained on a dataset of 2000 rotated shapes (see also Fig. 8).

(Scott & Serre, 1996), we then theoretically and empirically demonstrate the capacity of a model equipped with *distributed equivariant operators* in latent space to disentangle a range of affine image transformations including translations, rotations and combinations thereof.

## 2 EMPIRICAL LIMITATIONS OF TRADITIONAL DISENTANGLEMENT

In this section we empirically explore the limitations of traditional disentanglement approaches, in both unsupervised (variational autoencoder and variants) and supervised settings.

**VAE, beta-VAE and CCI-VAE**   We show that, consistent with results from prior literature, a variational autoencoder model (VAE) and its variants are successful at disentangling the factors of variation on a simple dataset. We train a VAE, beta-VAE and CCI-VAE (Kingma & Welling, 2014; Higgins et al., 2016; Burgess et al., 2018) on a dataset composed of a single class of MNIST digits (the "4s"), augmented with 10 evenly spaced rotations (all details of the models and datasets are in App. B). After training, we qualitatively assess the success of the models to disentangle the rotation transformation through traditional latent traversals: we feed an image of the test set to the network and obtain its corresponding latent representation. We then sweep a range of values for each latent dimension while freezing the other dimensions, obtaining a sequence of image reconstructions for each of these sweeps. We present in Fig. 1A examples of latent traversals along a single latent dimension, selected to be visually closest to a rotation (see Fig. 5 for latent traversals along all other latent dimensions). We find that all these models are mostly successful at the task of disentangling rotation for this simple dataset, in the sense that a sweep along a single dimension of the latent maps to diverse orientations of the test image.

We then show that on a slightly richer dataset (MNIST with all digits classes), a VAE model and its variants fail to disentangle shape from pose. We train all three models studied (VAE,

beta-VAE, CCI-VAE) on MNIST augmented with rotation, and find that all these models fail to disentangle rotation from other shape-related factors of variation (see Fig. 1B for the most visually compelling sweep and Fig. 6 for sweeps along all latent dimensions). We further quantify the failure of disentanglement by measuring the variance along each latent in response to a digit rotation, averaged over many digits (see Fig. 1C and details of analysis in App. E). We find that the information about the transformation is distributed across the latents, in contradiction with the conventional notion of disentanglement. One possibility would be that the direction of variance is confined to a subspace, but that this subspace is not aligned with any single latent. In order to discard this possibility, we carry a PCA-based analysis on the latent representation (Fig. 1D and App. E) and we show that the variance in latent representation corresponding to image rotation is not confined to a low-dimensional subspace.

**Supervised Disentanglement** We further explore the limitations of traditional disentanglement in a supervised framework. We train an autoencoder on pairs of input and target digit images (Fig. 1E), where the target image is a rotated version of the input image with a discrete rotation angle indexed by an integer value $k$. The input image is fed into the encoder to produce a latent representation. This latent representation is then multiplied by a matrix operator $\psi_k$, parameterized by the known transformation parameter $k$. This matrix operator, which we call the *disentangled operator*, is composed of a 2-by-2 diagonal block with a rotation matrix and an identity matrix along the other dimensions (shown in Fig. 1E). The disentangled operator (i) is consistent with the cyclic structure of the group of rotations and (ii) only operates on the first two latent dimensions, ensuring all other dimensions are invariant to the application of the operator. The transformed latent is then decoded and compared to the target image using an L2 loss (in addition, the untransformed latent is decoded and compared to the original image for regularization purposes). The only trainable parameters are the encoder and decoder weights. We use the same architecture for the encoder and decoder of this model that we use for the VAE models in the previous section. This supervised disentanglement model partly succeeds in mapping rotation to a single latent on rotated MNIST (Fig. 1E top row). However, there remains some digits for which disentanglement fails (Fig. 1E bottom row). ~~It is difficult to evaluate the capacity of the model to learn to rotate many different images with MNIST, because MNIST is only composed of 10 classes of shapes corresponding to the 10 different digits.~~ To further expose the limitations of this model, we design a custom dataset composed of 2000 simple shapes in all possible orientations. When trained on this extensive dataset, we find that ~~our~~ the model fails to capture rotations on many shapes. Instead, it replaces the shape of the input image with a mismatched stereotypical shape (Fig. 1F). We reproduce all these results with translation in the appendix (Fig. 8-12).

In conclusion, we find that common disentanglement methods are limited in their ability to disentangle pose from shape in a relatively simple dataset, even with strong supervision (see also Locatello et al. (2019b)). We cannot empirically discard the possibility that a larger model, trained for longer on even more examples of transformed shapes, could eventually learn to disentangle pose from shape. However, in the next section we will prove, using arguments from topology, that under the current definition of disentanglement, an autoencoder cannot possibly learn a perfect disentangled representation for all poses and shapes. In Sec. ~~4~~3.4 and Sec. 4, we will show that another type of model —inspired by group representation theory— can properly disentangle pose from shape.

## 3 REFRAMING DISENTANGLEMENT

In this section, we formally prove that traditional disentanglement by a continuous encoder is mathematically impossible for a large family of transformations, including all affine transformations. We then provide a more flexible definition of disentanglement that does not suffer from the same theoretical issues.

### 3.1 MATHEMATICAL IMPOSSIBILITY OF DISENTANGLEMENT

We first consider a simple example case where disentanglement is impossible. We consider the space of all images of 3 pixels $X = \mathbb{R}^3$, and the transformation acting on this space to be the group of integer finite translations, assuming periodic boundary conditions of the image in order to satisfy

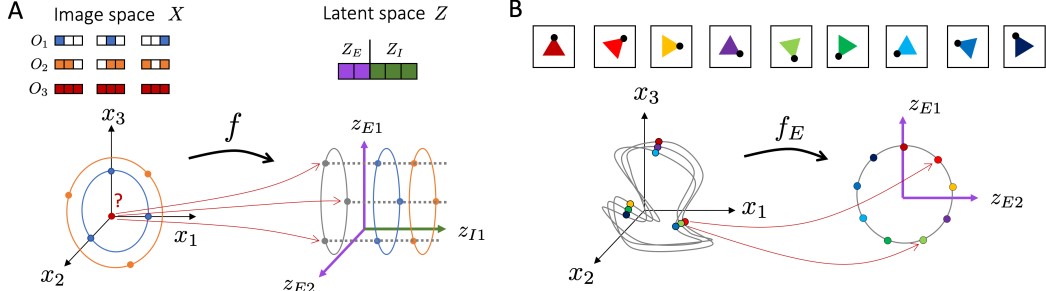

Figure 2: **Visual proof of the topological defects of disentanglement. A**. Top left: $O_1$, $O_2$ and $O_3$ are three examples of orbits of 3-pixel-images transformed by translation. Bottom: (left) orbits visualized in image space (points constitute the orbits, continuous lines are for visualization purposes); (right) orbits in latent space. When projected onto the equivariant subspace $Z_E$ (gray dotted lines), all orbits should collapse onto each other. Yet the orbit of a uniformly black image (red dot) contains a single point and thus cannot be mapped onto the other orbits. **B**. Discontinuity of $f_E$ around symmetric images. Top: consider an image of an equilateral triangle, with an infinitesimal perturbation on one corner (black dot), undergoing rotation (color changes are for visualisation purposes). Bottom: (left) after a rotation of 120°, the orbit in image space (here projected onto 3 dimensions for visualisation) almost loops back on itself; (right) in $Z_E$, each angle of rotation corresponds to a distinct point in space. Therefore, the encoder $f_E$ is discontinuous (as shown by the red arrows).

the group axiom of invertibility (Fig. 2A, see App. A for definitions). Given an image, the set of images resulting from the application of all possible translations to this image is called the *orbit* of this image. We note that the space of images $\mathbb{R}^3$ is composed of an infinite set of disjoint orbits. Can we find an encoder $f$ which maps every point of image space $X$ to a disentangled space $Z$?

To conform to the conventional definition of disentanglement (Higgins et al., 2018) (see App. C for a formal definition), $Z$ should be composed of two subspaces, namely (i) an equivariant subspace $Z_E$ containing *all and only* the information about the transformation (i.e. location along the orbit) and (ii) an invariant subspace $Z_I$, invariant to the transformation but containing all other information about the image (i.e. identity of the orbit). Each orbit should thus lie in a plane parallel to $Z_E$ (otherwise some information about the transformation would leak into $Z_I$), and all orbits projected onto $Z_E$ should map onto each other (otherwise some information about the identity of the orbit would leak into $Z_E$). We now consider the orbit containing the black image [0,0,0]. Since all translations of the black image are the black image itself, this orbit contains only one point. And yet, the image of this orbit in $Z_E$ should conform to the image of other orbits, which generally consist of 3 distinct points. Since a function cannot map a single point to 3 points, an encoder $f$ ensuring disentanglement for all images cannot exist.

Using similar topological arguments, we formally prove the following theorem (App. C.1), generalizing the observation above to a large family of transformations including translations, rotations and scalings.

**Theorem 1:** *Disentanglement into subspaces by a continuous encoder is impossible for any finite group acting on Euclidean space $\mathbb{R}^N$.*

## 3.2 Practical Examples of Topological Defects

The formal theorem of the previous section does not tell us how hard it would be to *approximate* disentanglement in practice. We show next that a disentangling encoder $f$ would need to be *discontinuous* around *all images that present a symmetry with respect to the transformation*, which makes this function very discontinuous in practice. As an example, we consider the image of an equilateral triangle undergoing rotation (Fig. 2B, color changes are for visualisation purposes). Due to the symmetries of the triangle, a rotation of 120° of this image returns the image itself. Now we consider the same image with an infinitesimal perturbation on one corner of the triangle, breaking the symmetry

of the image. A rotation of $120°$ of this perturbed image returns an image that is infinitesimally close to the original image. And yet the equivariant part of the encoder $f_E$ (i.e. the projection of $f$ onto the equivariant subspace $Z_E$) should map these two images to disjoint points in the equivariant subspace $Z_E$, in order to properly encode the rotation transformation. Generalizing this argument to all symmetric images, we see that a disentangling encoder would be discontinuous in the neighborhood of all images that present a symmetry with respect to the transformation to disentangle. This is incompatible with most deep learning frameworks, where the encoder is usually a neural network implementing a continuous function. We provide a formal proof of the discontinuity of $f_E$ in App. C.2.

The invariant encoder $f_I$ (i.e. the projection of $f$ onto the invariant subspace $Z_I$) also presents topological defects around symmetric images. We provide both a visual proof and a formal proof of these defects in App. C.2.

## 3.3 A ~~More~~ More Flexible Definition of Disentanglement

An underlying assumption behind the traditional definition of disentanglement is that the data is naturally acted upon by a set of transformations that are orthogonal to each other, and that modify well-separated aspects of the data samples. However, in many cases this separation between factors of variation of the data is not possible (as also noted by Higgins et al. (2018)). We notice that the current definition of disentanglement unnecessarily conflates the *objective* of isolating factors of variation with the *algorithm* of mapping these factors into distinct subspaces of the internal representation. In order to build a model that respects the structure of the data and its transformations, the latent space should instead *preserve the entanglement between factors of variation that are not independent*. ~~Thus, we~~

A model equipped with a latent operator is equivariant to a transformation if encoding a sample then applying the latent operator is equivalent to transforming the sample first then encoding. Formally, we say a model $f : X \to Z$ is equivariant to a transformation $g_k$ with parameter $k$ if for any input $x \in X$

$$f(\phi_k(x)) = \psi_k(f(x)), \forall k \in K, \tag{1}$$

where $K$ is the space of transformation parameters. The operators $\phi_k$ and $\psi_k$ capture how the transformation $g_k$ acts on the input space and representation space respectively. With this view, we turn to a definition of disentanglement in which the transformations are modelled as *distributed* operators (i.e. not restricted to a subspace) in the latent space.

~~Definition 1:~~

**Definition 1.** *A representation is disentangled with respect to a set of transformations, if there is a family of controllable operators, potentially acting on the entire representation, where each operator corresponds to the action of a single transformation and the resulting model is equivariant.*

These operators are ~~A representation is said to be disentangled with respect to a particular decomposition of a symmetry group into subgroups, if there is a family of known operators acting on this representation, potentially distributed across the full latent, where each operator is equivariant to the action of a single subgroup.~~controllable ~~This definition~~is in the sense that they have an explicit form, thus allowing the user to manipulate the latent representation by applying the operator. This definition, more flexible than traditional disentanglement in the choice of the latent operators, ~~while serving the same objective as traditional disentanglement (isolation and identification of~~obeys to the same *desiderata* of identification and isolation of the factors of variations present in the data~~).~~.

## 3.4 The Shift Operator for Affine Transformations

The flexibility of Def.1 unlocks a powerful toolbox for understanding and building disentangled models. Transformations in data often have additional structure describing how to (1) undo a transformation (invertibility), (2) leave a sample unchanged (identity), and (3) rearrange parenthesis for the order transformations are applied (associativity). A collection of transformations with a rule for combining two transformations into another satisfying these three simple requirements has a group structure (see App. A for a formal definition). With a group structure, we can decompose transformations into subgroups, describe how to represent transformations as matrices, and build

flexible disentangled models.

Using classical results from the linear representation of finite groups (Scott & Serre, 1996), we show ~~in App. C.3~~ now that a carefully chosen distributed operator in latent space, —the shift operator $\psi_k$ (shown in Fig. 3A)— is linearly isomorphic to specific transformations that include integer pixel translations and rotations. With this operator, we can learn a latent space equivariant to any affine transformation using a simple *linear* autoencoder. Consider a linear encoder model $f = W$. If we want $W$ to be an equivariant invertible linear mapping $W$ between $X$ and $Z$, Equation 1 rewrites as follows:

$$W \phi_k(x) = \psi_k(W\,x) \ \forall x \in X(= \mathbb{R}^N), \forall k \in K \tag{2}$$

where $\phi_k$ and $\psi_k$ are the representations of $g_k \in G$ on the image and latent space respectively, as defined in App. A. For $W$ to be equivariant, Equation 2 must be true for every image $x$. As $W$ is invertible, Equation 2 is true if and only if, $\forall k \in K$, the two representations $\psi_k$ and $\phi_k$ are isomorphic:

$$\forall k \in K, \phi_k = W^{-1}\,\psi_k\,W \tag{3}$$

We consider additional properties on $\phi$ corresponding to the assumptions (i) that $G$ is cyclic of order $K$ with generator $g_0$ and (ii) $\phi$ is isomorphic to the regular representation of $G$ (see Scott & Serre (1996)). These properties are respected by all cyclic linear transformations of finite order $K$ of the images (see App. A for definitions)~~. Special cases of these transformations include all discrete and cyclic affine transformations~~, such as integer pixel translation with periodic boundary conditions, or rotations. ~~A practical consequence of this proof is that it is possible to learn a latent space equivariant to any affine transformation with this operator, using a simple *linear* autoencoder.~~

**~~Definition and properties of the shift operator~~** ~~For a transformation $g_k \in G$ such that $g_k = g_0^k$, where $g_0$ is the generator of~~ Given that the encoder and decoder are linear and invertible, the two representations $\phi$ and $\psi$ must be isomorphic. Two representations are isomorphic if and only if they have the same character (Scott & Serre, 1996, Theorem 4, Corollary 2) (see App. A for a definition of characters). We thus want to choose $\psi$ such that it preserves the character of the representation $\phi$ corresponding to the action of $G$ ~~, the corresponding shift operator $\psi_k$ is the unitary shift matrix exponentiated by $k$. Note that this operator is distributed as it acts~~ on the dataset of images. Importantly, we will see that our proposed operator needs to be *distributed* in the sense that it should act on the full latent ~~space, contrary to conventional disentanglement models.Importantly, the proposed shift operator is flexible in the sense that it does not require specific knowledge of which transformation groupis applied to the data (i. e. for example rotations or translations), but only the order of the group and the *character* of its representation (App. A). Furthermore, in App.~~ code.

Let us consider the matrix of order $K = |G|$ that corresponds to a shift of elements in a $K$-dimensional vector by $k$ positions. We construct from $M_k$ the *shift operator* as a representation of the group's action on the latent space. For each $g_k \in G$ its corresponding shift operator is the block diagonal matrix of order $N$ composed of $\frac{N}{K}$ repetition of $M^k$.

$$M_k := \begin{bmatrix} 0 & 0 & \dots & 1 \\ 1 & 0 & \dots & 0 \\ 0 & 1 & 0 & \vdots \\ \vdots & & & \\ 0 & \dots & 1 & 0 \end{bmatrix}^k \tag{4} \qquad \psi_k := \begin{bmatrix} M^k & & & \\ & M^k & & \\ & & \dots & \\ & & & M^k \end{bmatrix} \tag{5}$$

We show in App. C.3 that the shift operator has the same character as the representation $\phi$ corresponding to the action of $G$ on the dataset of images. Thus, the two are isomorphic and using this shift operator ensures that an equivariant invertible linear mapping $W$ exists between image space and the latent space equipped with the shift operator. In Appendix C.3.2, we ~~also propose a complex version of the shift operator~~ show that we can also replace this shift operator by

a complex diagonal operator, which is more computationally efficient ~~. Although these operators cannot theoretically be mapped to continuous transforms, we note that any continuous transform can be approximated using finite groups. In the remainder of the paper, we denote $\psi_{t,k,N}$ the shift operator that corresponds to $t \in T$ where $T$ is a cyclic group of transformations (e. g. rotations) of order $N$ with generator $t_0$ and order $K$, such that $t = t_0^k$.~~

~~In the next section, we show how these theoretical results can lead to practical and effective disentangling models for affine transformations such as rotations and translations.~~ to multiply with the latent.

The shift operator computes a shift of the latent space and we use this form of operator to represent any finite cyclic group of affine transformations (i.e. either rotation, translation in x, translation in y). The role of the encoder is to construct a latent space where transformations can be represented as shifts. Importantly, the shift operator **does not require knowledge of the transformation in advance**, only the cycle order of each group, which is an assumption we will relax with the weakly supervised version of the shift operator. These assumptions allow us to guarantee a linear equivariant model can be learned with pairs of examples (see our training objectives in App. B.2). Note that the shift operator only handles cyclic groups with finite order (or a product of such groups). In order to tackle continuous transformations, a discretisation step could be added and we leave the exploration of this extension for future work.

## 4 DISTRIBUTED DISENTANGLEMENT IN PRACTICE

Our empirical (Sec. 2) and theoretical (Sec. 3) findings converge to show the difficulties of disentangling *even simple affine transformations* into distinct subspaces. Here we show that, using distributed equivariant operators instead, it is practically possible to learn to disentangle these affine transformations, according to our more flexible definition of disentanglement.

### 4.1 THE SUPERVISED SHIFT OPERATOR MODEL

Guided by our theoretical results, we train a supervised non-variational autoencoder using pairs of samples and their transformed version (with a known transformation indexed by $k$) using the distributed shift operator from Sec. 3.4 (shown in Fig. 3A) instead of the disentangled operator from Sec. 2. We feed the original sample $x$ to a linear invertible [1] (or quasi-invertible, see App. B.2) encoder that produces a latent representation. The latent representation is then multiplied by the shift operator matrix parametrized by $k$. The transformed latent is then decoded and L2 loss between the two reconstructions ($x$ and its transformed version) and their respective ground-truth images is back-propagated.

As predicted by character theory, our proposed model is able to correctly structure the latent space, such that applying the shift operator to the latent code at test time emulates the learned transformation (see Fig. 3A~~and~~, test MSE reported in Table 2~~). Also consistent with~~, and the LSBD disentanglement measure from Anonymous (2021) and reported in App. E.1.1). Interestingly, test MSE for rotations are lower than for translations. We believe this is due to the fact that in the case of translations, changes in the image induced by each transformation are less visually striking than with rotations. Consistent with the theory, the same linear autoencoder equipped with a *disentangled* operator fails at learning the transformation (Fig. 3B). We reproduce these results for translation in App. E. We thus show that, unlike prior approaches, our theoretically motivated model is able to ~~learn to~~ disentangle affine transformations from examples.

### 4.2 THE WEAKLY SUPERVISED SHIFT OPERATOR MODEL

Here we show that our method can also learn to disentangle transformations in a *weakly supervised setting* where the model is *not* given the transformation parameter between pairs of transformed images (e.g. rotation angle) during training. We consider the case of a single transformation for simplicity. We encode samples by pairs $x_1, x_2$ (with $x_2$ a transformed version of $x_1$) into $z_1$ and $z_2$ respectively, and use a phase correlation technique to identify the shift between $z_1$ and $z_2$, as in

---

[1]~~or quasi-invertible, see App. B.2~~

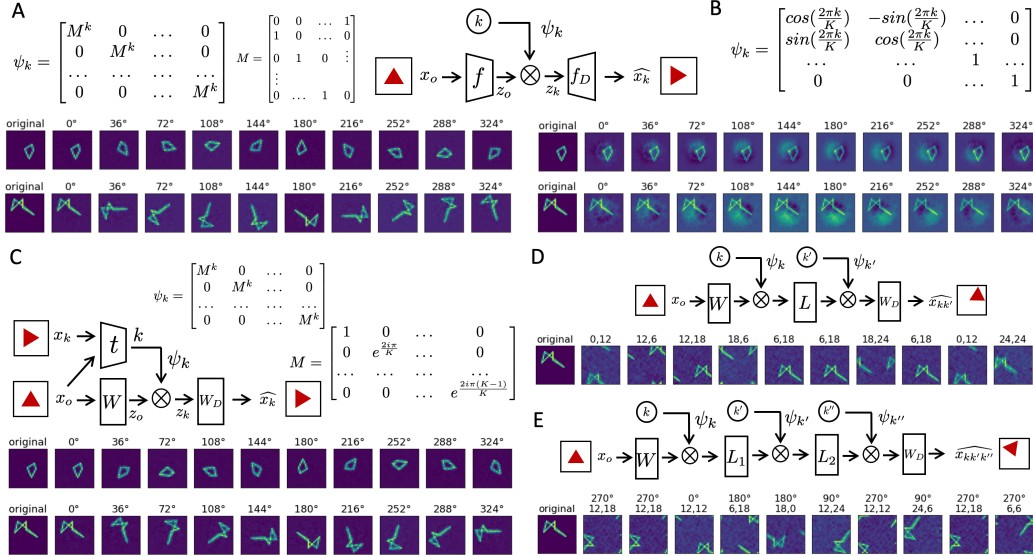

Figure 3: **Success and flexibility of proposed distributed shift operator models**: **A.** Proposed shift operator model successfully learns rotation on simple shapes. **B.** Disentangled operator fails to learn rotation. **C.** Weakly supervised shift operator model, using the complex version of the shift operator, successfully rotates simple shapes. Note that the model maps ground-truth counter clockwise rotations to clockwise rotations, while respecting the cyclic structure of the group. **D.** Stacked shift operator model succeeds on conjunction of translations. **E.** Stacked shift operator model succeeds on conjunction of translations and rotations. Numbers above plots indicate rotation angle and/or translation in $x$ and $y$ respectively.

Reddy & Chatterji (1996). An L2 loss is applied on reconstructed samples, with the original sample transformed according to all possible $k$ and weighted according to soft-max scores given by the cross-correlation method (see App. B.3 for details). Here and in the remainder of the experiments, we use the complex version of the shift operator for computational efficiency (shown in Fig. 3C). This weakly supervised version of the model has an extra free parameter which is the number of latent transformations, not known a priori. Let us denote $K_L$ this number, which can be different than the ground-truth order of the group $K$. We explore the effect of different $K_L$ in App. B.3. The results of this model (with 10 latent transformations $K_L$) are shown in Fig. 3C. The weakly supervised shift operator model works almost as well as its supervised counterpart, and this is confirmed by test MSE (see Table 2). The same model can successfully be trained on MNIST digits (Fig. 14).

### 4.3 MULTIPLE TRANSFORMATIONS: STACKING SHIFT OPERATORS

So far, we have only considered the case of a single type of transformation at a time. When working with real images, there are more than one type of transformation jointly acting on the data, for example a rotation followed by a translation. Here we show how we can adapt our proposed shift operator model to the case of multiple transformations.

**Stacked shift operator model** In the case of multiple transformations, a group element is a composition of consecutive single transformations. For example, elements of the Special Euclidean group (i.e. translations and rotations) are a composition of the form $a_y \, a_x \, h$, where $a_x$ is an element of the $x$-translation subgroup, $a_y$ of the $y$-translation subgroup, and $h$ of the image rotation subgroup. Our theory in C.3 ensures that each of these subgroup's action in image space is linearly isomorphic to the repeated regular representation $\psi$. We can thus match the structure of the learning problem by simply stacking linear layers and operators. We build a stacked version of our shift operator model, in which we use one complex shift operator for each type of transformation, and apply these operators to the latent code in a sequential manner, akin to Tai et al. (2019). Specifically,

consider an image $x \in X$ that encounters a consecutive set of transformations $g_n, g_{n-1}, \ldots g_1$, with $g_i \in G_i \ \forall i$. We encode $x$ into $z$ with a linear invertible encoder $z = W \ x$. We then apply the operator on the latent space, $\psi_1(g_1)$, corresponding to the representation of $g_1$ on the latent space $Z$. We then apply a linear layer $L_1$ before using the operator corresponding to $G_2$. The resulting latent code after all operators have been applied is $z' = \psi_n(g_n)L_{n-1} \ldots \psi_2(g_2)L_1\psi_1 z$. The transformed latent code $z'$ is fed to the linear decoder, in order to produce a reconstruction that will be compared with the ground-truth transformed image $x'$, as in the single transformation case.

**Translations in X and Y** Consider the conjunction of translations in $x$ and $y$ axes of the image. This is a finite group of 2D translations. This group is a direct product of two cyclic groups, and it is abelian (i.e. *commutative*). We refer the interested reader to App. D.1 for details on direct products. To tackle this case with the stacked shift operator model, we first use the shift operator $\psi_{x,k,N}$ corresponding to the translation in $x$, then apply a linear layer denoted $L_1$, before using the operator $\psi_{y,k',N}$ corresponding to the translation in $y$: $z' = \psi_{y,k',N}L_1\psi_{x,k,N} z$. We train this stacked model on translated shapes with 5 integer translations in both $x$ and $y$ axes (i.e. the group order is 25). Results reported in Fig. 3D show that the stacked shift operator model is able to correctly handle the group of 2D translations.

**Translations and rotations** We consider a discrete and finite version of the Special Euclidean group, where $A$ is a finite group of 2D translations presented in the previous section and $H$ a finite cyclic group of rotations. This group has a semi-direct product structure (see App. D.3 for details) and is *non-commutative*, contrary to the 2D translations case. With the stacked shift operators model, we first use the operator $\psi_{h,j,N}$ corresponding to the rotation $h$, then the one for translation in $x$, then the one for $y$-translation. The resulting transformed latent code is $z' = \psi_{y,k',N}L_2\psi_{x,k,N}L_1\psi_{h,j,N} z$. We train this stacked model on discrete rotations followed by integer translations and discrete rotations, using 5 integer translations in both $x$ and $y$ axes and 4 rotations. Results reported in Fig. 3E and MSE Table 2 show that the model is perfectly able to structure the latent space such that the group structure is respected. Additionally, Figures 17 and 18 show pairs of samples and the reconstructions by the stacked shift operator model in the cases of (i) translation in both x and y axes and (ii) rotations and translations in both axes. In appendix Figure 16 we also explore the case where the order of the group of rotations is 5, breaking the semi-direct product structure (see note in Appendix D.4.1) and show that the stacked shift operator nonetheless performs with great performance.

**Insight from representation theory on the structure of hidden layers** When dealing with multiple transformations (e.g. rotations and translations), we know the form of the operator for every subgroup (shift operator), but we do not know *a priori* the form of the resulting operator for the entire group. In App. D.1 and D.4 we derive from representation theory the operator for the entire group in the 2D translation and Special Euclidean group cases and show that they can be built from each subgroup's operator in a non-trivial way. Importantly, we show that the resulting operator for the discrete finite Special Euclidean case has a block matrix form representation based on representations of both translations and rotations. This is expected: this group is non-commutative, so the correct operator *cannot* be diagonal otherwise two operators corresponding to two elements would commute. Equipped with this theory we can derive insights about the form that intermediate layers should take after training. In particular, we show (App. D.2) that the layer $L_1$ should be a block diagonal matrix consisting of repetitions of a permutation matrix that reorders elements in $\psi_{x,k,N} z$. Similarly, in the case of translations and rotations together, $L_2$ must reorder $\psi_{y,k',N}$, and $L_1$ be the product of two matrices $L_1 = PQ$, where $Q$ is a $N$ by $N$ block diagonal matrix and $P$ is reordering the rows of the vector $Q\psi_{h,j,N}z$ (see App. D.5). In future work, we plan to explore the use of these insights to regularize internal layers of stacked shift operator models.

## 5 DISCUSSION

Finding representations that are equivariant to transformations present in data is a daunting problem with no single solution. A large body of work (Cohen et al., 2020; 2018; Esteves et al., 2018; Greydanus et al., 2019; Romero et al., 2020; Finzi et al., 2020; Tai et al., 2019) proposes to hard-code equivariances in the neural architecture, which requires *a priori* knowledge of the transformations present in the data. In another line of work, Falorsi et al. (2018); Davidson et al. (2018); Falorsi et al.

(2019) show that the topology of the data manifold should be preserved by the latent representation, but these studies do not address the problem of disentanglement. Higgins et al. (2018) have proposed the framework of disentanglement as equivariance that we build upon here. Our work extends their original contribution in multiple ways. First, we show that traditional disentanglement introduces topological defects (i.e. discontinuities in the encoder), even in the case of simple affine transformations. Second, we conceptually reframe disentanglement, allowing equivariant operators to act on the entire latent space, so as to resolve these topological defects. Finally, we show that models equipped with such operators successfully learn to disentangle simple affine transformations.

An important direction for future work will be to expand the reach of the theory to a broader family of transformations. In particular, it is unclear how the proposed approach should be adapted to learn transformations which are not affine or linear in image space, such as local deformations, compositional transformations (acting on different objects present in an image), and out-of plane rotations of objects in images (but see Dupont et al. (2020) for an empirical success using a variant of the shift operator). Another important direction would be to extend the theory and proposed models to continous Lie groups. Moreover, our current implementation of disentanglement relies on some supervision, by including pairs of transformed images~~(with or without knowledge of the parameter of the transformation occurring between them). It~~, moreover it would be important to understand how disentangled representations can be learned without such pairs ~~of transformed images~~ (see Anselmi et al. (2019); Zhou et al. (2020) for relevant work).

Finally, our work lays a theoretical foundation for the recent success of a new family of methods that —instead of enforcing disentangled representations to be restricted to distinct subspaces— use operators (hard-coded or learned) acting on the *entire* latent space (Connor & Rozell, 2020; Connor et al., 2020; Dupont et al., 2020; Giannone et al., 2020; Quessard et al., 2020) (see also (Memisevic & Hinton, 2010; Cohen & Welling, 2014; Sohl-Dickstein et al., 2017) for precursor methods). These methods work well where traditional disentanglement methods fail: for instance by learning to generate full 360° in-plane rotations of MNIST digits (Connor & Rozell, 2020), and even out-of-plane rotations of 3D objects (Dupont et al., 2020). ~~Unlike our simple theoretically-derived models which are suited only for affine transformations, these~~ These methods use distributed operators in combination with *non-linear* autoencoder architectures, an interesting direction for future theoretical investigations. Moreover, in the case where the latent operators cannot be determined in advance like in the affine case, these operators could be learned like in Connor et al. A benefit of this approach is that multiple operators can be learned in the same subspace, instead of the stacking strategy that we needed to use in the case of hard-coded shift operators.

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

# Appendix

# Table of Contents

## A   PREREQUISITES IN GROUP THEORY AND REPRESENTATION THEORY

**Definition of a group**   A group is a set $G$ together with an operation $\circ : G \times G \to G$ such that it respects the following axioms:

- Associativity: $(g_k \circ g_{k'}) \circ g_{k''} = g_k \circ (g_{k'} \circ g_{k''})$ with $g_k, g_{k'}, g_{k''} \in G$.
- Identity element: there exists an element $e_G \in G$ such that, for every $g_k \in G$, $g_k \circ e_G = e_G \circ g_k = g_k$, and $e_G$ is unique.
- Inverse element: for each $g_k \in G$, there exists an element $g_{k'} \in G$, denoted $g_k^{-1}$ such that $g_k \circ g_k^{-1} = g_k^{-1} \circ g_k = e_G$.

In the paper, for clarity we do not write explicitly the operation $\circ$ unless needed.

**Finite cyclic groups**   We will be interested in *finite* groups, composed of a finite number of elements (i.e. the order of $G$).

A *cyclic* group is a special type of group that is generated by a single element, called the generator $g_0$, such that each element can be obtained by repeatedly applying the group operation $\circ$ to $g_0$ or its inverse. Every element of a cyclic group can thus be written as $g^k = g_0^k$. Note that every cyclic group is abelian (i.e. its elements commute).

A group $G$ that is both finite and cyclic has a finite order $K$ such that $g_0^K = e_G$.

**Representation and equivariance** Informally, for a model to be equivariant to a group of transformations means that if we encode a sample, and transform the code, we get the same result as encoding the transformed sample. (Higgins et al., 2018) show that disentanglement can be viewed as a special case of equivariance, where the transformation of the code is restricted to a subspace. We provide a formal definition of equivariance below after introducing notations.

In the framework of group theory, we consider $\phi$ a **linear representation** of the group $G$ acting on $X$ (Scott & Serre, 1996): $\phi : G \to GL(X)$. Each element of the group $g_k \in G$ is represented by a matrix $\phi(g_k) = \phi_k$, and $\phi_k$ is a matrix with specific properties:

1. $\phi$ is a homomorphism: $\phi(g_k g_{k'}) = \phi(g_k)\phi(g_{k'})$, $g_k, g_{k'} \in G$.

2. $\phi_k$ is invertible and $\phi_k^{-1} = \phi(g_k^{-1})$ as $\phi(g_k^{-1})\phi(g_k) = \phi(g_k g_k^{-1}) = \phi(e_G) = I$. where $I$ is the identity matrix in $GL(X)$.

The set of matrices $\phi_k$ form a *linear representation* of $G$ and they multiply with the vectors in $X$ as follows:

$$\phi_k : X(= \mathbb{R}^N) \to X \ s.t. \ \forall x \in X, \phi_k(x) \in X. \tag{6}$$

The **character** of $\phi$ is the function $\chi_\phi$ such that for each $g_k \in G$, it returns the trace of $\phi(g_k) = \phi_k$, i.e. $\chi_\phi(g_k) = Tr(\phi_k)$. Importantly, the character of a representation completely characterizes the representation up to a linear isomorphism (i.e. change of basis) (Scott & Serre, 1996). The **character table** of a representation is composed of the values of the character evaluated at each element of the group.

Similarly, we denote a linear representation of the action of $G$ onto the latent space $Z$ by $\psi : G \to GL(Z)$ such that $\forall k, \ \psi(g_k) = \psi_k$. While corresponding to the action of the same group element $g_k \in G$, $\psi_k$ does not have to be the same as $\phi_k$, as it represents the action of $G$ on $Z$ and not on $X$.

$$\psi_k : \ Z(= \ \mathbb{C}^N) \to Z \ s.t. \ \forall z \in Z, \psi_k(z) \in Z. \tag{7}$$

Note that we consider a complex latent space. With these notions, formally the model $f : X \to Z$ is **equivariant** to the group of transformations $G$ that acts on the data if, for all elements of the group $g_k$ and its action $\phi_k$ and $\psi_k$ on the spaces $X$ and $Z$ respectively, we have:

$$\forall x \in X, \forall k \in K, f(\phi_k(x)) = \psi_k(f(x)) \tag{8}$$

Finally, a group action on image space $\mathbb{R}^N$ is said to be **affine** if it affects the image through an affine change of cooordinates.

## B  EXPERIMENTAL DETAILS

### B.1  DATASET GENERATION

**Simple shapes** We construct a dataset of randomly generated simple shapes, consisting of 5 randomly chosen points connected by straight lines with 28x28 pixels. We normalize pixel values to ensure they lie within $[0, 1]$. For all experiments, we use a dataset with 2000 shapes. For each shape, we apply 10 counterclockwise rotations by $\{0°, 36°, 72°, \dots, 324°\}$ using scikit-image's rotation functionality (see van der Walt et al. (2014)). For translations along the x-axis or y-axis we apply 10 translations using numpy.roll (see Van Der Walt et al. (2011)). This ensures periodic boundary conditions such that once a pixels is shifted beyond one edge of the frame, it reappears on the other edge. For experiments with supervision involving pairs, we construct every possible combination of pairs, $x_1, x_2$ and apply every transformation to both $x_1$ and $x_2$. For datasets containing multiple transformations, we first rotate, then translate along the x-axis, then translate along the y-axis. We use 50% train-test split then further split the training set into 20%-80% for validation and training.

**MNIST (LeCun et al., 2010)** Similar to simple shapes, we construct rotated and translated versions of MNIST by applying the same transformations to the original MNIST digits. We normalize

pixel values to lie between $[0, 1]$. For supervised experiments, we similarly construct every combination of pairs $x_1, x_2$ and apply transformations to both $x_1$ and $x_2$. Since constructing every combination of transformed pairs would lead to many multiples the size of the original MNIST, we randomly sample from the training set to match the original number of samples. We use the full test set augmented with transformations for reporting losses.

## B.2 MODEL ARCHITECTURES AND TRAINING

We implement all models using PyTorch with the Adam optimizer (Paszke et al., 2019; Kingma & Ba, 2014).

**Variational Autoencoder and Variants** We implement existing state-of-the-art disentanglement methods $\beta$-VAE and CCI-VAE, which aim to learn factorized representations corresponding to factors of variation in the data. In our case the factors of variation are the image content and the transformation used (rotation or translations). We use the loss from CCI-VAE made up of a reconstruction mean squared error and a Kullback–Leibler divergence scaled by $\beta$:

$$L = \sum_i^m (x_i - f_D(f(x_1)))^2/m + \beta|KL(q(z|x), p(z)) - C| \tag{9}$$

where $m$ is the number of samples and Kullback-Leibler divergence is estimated as $KL(q(z|x), p(z)) = -0.5 * \sum_i^d (1 + \ln(\sigma_i^2) - \mu_i^2 - \sigma_i^2)$, where $d$ is the latent dimension (see Kingma & Welling (2014)). For a standard VAE, we use $C = 0$ and $\beta = 1.0$. For $\beta$-VAE, we sweep over choices of $\beta$ with $C = 0$. For CCI-VAE, we sweep over choices of $\beta$ and linearly increase $C$ throughout training from $C = 0$ to $C = 36.0$.

We use the encoder/decoder architectures from Burgess et al. (2018) comprised of 4 convolutional layers, each with 28 channels, 4x4 kernels, and a stride of 2. This is followed by 2 fully 256-unit connected layers. We apply ReLU activation after each layer. The latent distribution is generated from 30 units: 15 units for the mean and 15 units for the log-variance of a Gaussian distribution. The decoder is comprised of the same transposed architecture as the encoder with a final sigmoid activation.

**Autoencoder with latent operators** For the standard autoencoder and autoencoders with the shift/disentangled latent operators, we use supervised training with pairs $(x_1, x_2)$ and a transformation parameter $k$ corresponding to the transformation between $x_1$ and $x_2$. The loss is the sum of reconstruction losses for $x_1$ and $x_2$:

$$L = \sum_{i=1}^m (x_{1,i} - f_D(f(x_{1,i})))^2/m + \sum_{i=1}^m (x_{2,i} - f_D(\psi_k(f(x_{1,i}))))^2/m \tag{10}$$

where $m$ is the number of samples and $\psi$ is the disentangled or shift latent operator. For a standard autoencoder, only the first reconstruction term is present in the loss function.

For the non-linear autoencoders, we use the same architecture above based on CCI-VAE. In the linear case, we use a single fully 28x28 connected layer and an 800 dimensional latent space. We use 800 dimensions to approximate the number of pixels (28x28) and ensure that $K=10$ divides the latent dimension. This is an approximation of the correct theoretical operator (which should be invertible) that works well in practice.

**Weakly supervised shift operator model** We use linear encoders and decoders with a 784 dimensional latent space to match the number of pixels. Indeed the weakly supervised shift operator uses the complex version of the shift operator, so we can perfectly match the size of the image. Training is done with L2 loss on all possible reconstructions of $x_1$ (of each training pair $i$), weighted by scores $\alpha_{i,k}$. Appendix B.3 below gives a detailed explanation of the computation of the scores $\alpha_{i,k}$:

$$L = \sum_{i=1}^m (x_{1,i} - f_D(f(x_{1,i})))^2/m + \sum_{i=1}^m \sum_{k=1}^{K_L} \alpha_{i,k} (x_{2,i} - f_D(\psi_k(f(x_{1,i,k}))))^2/m \tag{11}$$

where $m$ is the number of samples. At test-time, we use the transformation with maximum score $\alpha_{i,k}$.

**Stacked shift operator model** We use linear encoders and decoders with a 784 dimensional latent space to match pixel size, as we use in the stacked model the complex version of the shift operator. Intermediate layers $L_i$ are invertible linear layers of size 784 as well. Training is done with L2 loss on reconstructed samples as in the autoencoder with shift latent operator (see Equation 10.)

### B.3 WEAKLY SUPERVISED SHIFT OPERATOR TRAINING PROCEDURE

**Method** We experimentally show in Section 4.2 that the proposed operator $\psi_k$ works well in practice. Additionally, we developed a method for inferring the parameter of the transformation $k$ that needs to act on the sample. We encode samples by pairs $x_1, x_2$ (with $x_2$ a transformed version of $x_1$) into $z_1$ and $z_2$ respectively, and use a classical phase correlation technique (Reddy & Chatterji, 1996) to identify the shift between $z_1$ and $z_2$, described below. Then, we use the complex diagonal shift operator parametrized by the inferred transformation parameter $k$.

Importantly, in the weakly supervised version of the shift operator, the model has an extra free parameter which is the number of latent transformations, not known a priori. Let us denote $K_L$ this number, which can be different than the ground-truth order of the group $K$.

To infer the latent transformation that appear between $z_1$ and $z_2$, we compute the cross-power spectrum between the two codes $z_1$ and $z_2$, that are both complex vectors of size $N$ and obtain a complex vector of size $N$. We repeat $K_L$ times this vector, obtaining a $K_L$ x $N$ matrix, of which we compute the inverse Fourier transform. The resulting matrix should have rows that are approximately 0, except at the row $k$ corresponding to the shift between the two images, see Reddy & Chatterji (1996). Thus, we compute the mean of the frequencies of the real part of the inverse Fourier result (i.e. the mean over the $N$ values in the second dimension). This gives us a $K_L$-dimensional vector, which we use as a vector of scores of each $k$ to be the correct shift between $z_1$ and $z_2$. During training, we compute the soft-max of these scores with a temperature parameter $k$, this gives us $K_L$ weights $\alpha_k$. We transform $z_1$ with all $K$ possible shift operators, decode into $K$ reconstructions $x_k$, and weight the mean square error between $x_2$ and each $x_k$ by $\alpha_k$ before back-propagating. This results, for each samples pair $(x_{1,i}, x_{2,i})$, in the loss:

$$L = \sum_{i=1}^{m}(x_{1,i} - f_D(f(x_{1,i}))^2/m + \sum_{i=1}^{m}\sum_{k=1}^{K_L}\alpha_{i,k}(x_{2,i} - f_D(\psi_k(f(x_{1,i,k}))))^2/m \qquad (12)$$

where $m$ is the number of samples and $\alpha_{i,k}$ the scores for the pair $(x_{1,i}, x_{2,i})$. At test-time, we use the transformation with maximum score $\alpha_{i,k}$.

| Dataset | $K_L$ | |
| --- | --- | --- |
| | 10 | 21 |
| Shapes (10,0,0) | $0.001 \pm 0.0013$ | $0.0005 \pm 0.0001$ |
| Shapes (0,10,0) | $0.0097 \pm 0.0052$ | $0.0038 \pm 0.0013$ |
| Shapes (0,0,10) | $0.0115 \pm 0.0049$ | $0.005 \pm 0.0033$ |
| MNIST (10,0,0) | $0.0035 \pm 0.0039$ | $0.0074 \pm 0.0083$ |

Table 1: Comparing test mean square error (MSE) $\pm$ standard deviation of the mean over random seeds for different $K_L$. Numbers in () refer to the number of rotations, the number of translations on the $x$-axis, and the number of translations on the $y$-axis respectively.

**Effect of the number of latent transformations** In the weakly supervised shift operator model, the number of latent transformations $K_L$ is a free parameter. Interestingly, when using rotations the best cross-validated number of transformations is 10, which matches the ground-truth order of the group. For translations (either on the $x$ or the $y$ axis), best results are obtained using $K_L = 21$ which is larger than the ground-truth order of the group $K$. Table 1 compares test MSE for both values of $K_L$. We think that in the case of translations, changes in the image induced by each shift (each transformation) are less visually striking than with rotations, and a larger $K_L$ gives extra

flexibility to the model to identify the group elements and respects the group structure (namely its cyclic aspect).

### B.4 HYPER-PARAMETERS

**General hyper-parameters**  We sweep across several sets of hyper-parameters for our experiments. We report results for the model with the lowest validation test loss. To avoid over-fitting, results for any given model are also stored during training for the parameters yielding the lowest validation loss.

For experiments with simple shapes we sweep across combinations of

- 5 seeds: 0, 10, 20, 30, 40
- 4 batch sizes: 4,8,16,32
- 2 learning rates: 0.0005, 0.001

For MNIST, we sweep across combinations of

- 5 seeds: 0, 10, 20, 30, 40
- 4 batch sizes: 8,16,32, 64
- 2 learning rates: 0.0005, 0.001

In addition to these general parameters, we also sweep across choices of $\beta$, $\{4, 10, 100, 1000\}$ and latent dimension $\{10, 30\}$ for the variational autoencoder models and variants. We repeat all experiments across four seeds used to initialize random number generation and weight initialization.

**Weakly supervised shift operator hyper-parameters**  We perform a sweep over hyper-parameters as described above. Additionally for the weakly supervised model, we sweep over temperature $\tau$ of the soft-max that shapes the scores of each transformation over values $\tau = \{0.01, 0.1, 1.0\}$, and the number of transformations composing the operator family (i.e. the order of the group) over values 10 and 21, where 10 is the ground-truth order of the group.

**Stacked shift operator model hyper-parameters**  We perform a sweep over hyper-parameters as described above. The only exception is that for the case of the Special Euclidean group, we train only for 5 epochs, and try batch sizes $\{32, 64\}$ for MNIST and $\{16, 32\}$ for simple shapes, as the number of generated samples is high. Similarly, for the case of 2D translations, we use batch sizes $\{16, 32\}$ for simple shapes.

## C  REFRAMING DISENTANGLEMENT: FORMAL PROOFS

We consider the action of a finite group on image space $\mathbb{R}^N$. Using tools from topology, we show that it is impossible to learn a representation which disentangles the action of this group with a continuous encoder $f$.

### C.1  TOPOLOGICAL PROOF AGAINST DISENTANGLEMENT

We consider a finite group $G$ of cardinal $|G|$ that acts on $\mathbb{R}^N$. Given an image $x \in \mathbb{R}^N$, an orbit containing that image is given by $\{g_1 x, g_2 x, ..., g_K x\}$.

We consider an encoder $f : \mathbb{R}^N \to M$ that disentangles the group action. The image of $f$ is composed of an equivariant subspace and an invariant subspace. We define $f_E : \mathbb{R}^N \to M_E$ the projection of $f$ on its equivariant subspace and $f_I : \mathbb{R}^N \to M_I$ the projection of $f$ on its invariant subspace.

$f_E$ is equivariant to the group action. In equations:

$$\forall x, \forall g, f_E(gx) = g f_E(x) \tag{13}$$

For the disentanglement to be complete, $f_E$ should not contain any information about the identity of the orbit the image belongs to. If $O_1$ and $O_2$ are two distinct orbits, we thus have:

$$\forall x_1 \in O_1, \forall x_2 \in O_2, \exists g \in G, f_E(x_1) = g f_E(x_2) \tag{14}$$

$f_I$ is invariant to the group action:

$$\forall x, \forall g, f_I(gx) = f_I(x) \tag{15}$$

We also assume that the representation contains all the information needed to reconstruct the input image:

$$\forall x_1 \in O_1, \forall x_2 \in O_2, f_I(x_1) \neq f_I(x_2) \tag{16}$$

This last assumption corresponds to assuming that every image can be perfectly identified from its latent representation by a decoder (i.e. perfect autoencoder). We also assume that both the encoder and decoder are continuous functions. This is a reasonable assumption as most deep network architectures are differentiable and thus continuous. In the language of topology, the encoder $f$ is a *homeomorphism*, a continuous invertible function whose inverse is also a continuous function. Also called a topological isomorphism, an homeomorphism is a function that preserves all topological properties of its input space (see Munkres (2014) p.105). Here we prove that $f$ cannot preserve the topology of $\mathbb{R}^N$ while disentangling the group action of $G$.

Consider $f_E|_O$ the restriction of $f_E$ to a single orbit of an image $x$ without any particular symmetry. $f_E|_O$ inherits continuity from $f$, and it can easily be shown that $f_E|_O$ is invertible (otherwise, information about the transformation on this orbit is irremediably lost, and $f$ can thus not be invertible).

$f_E|_O$ is thus also a homeomorphism, and so it preserves the topology of the orbit $O$ in image space, which is a set of $|G|$ disconnected points.

By equation 14, we know that restrictions of $f_E$ to all other orbits have an image contained in the same topological space. As a consequence, the image of $f_E$ itself is a set of $|G|$ disconnected points.

Since $f_E$ is a projection of $f$, the image of $f$ should at least be composed of $|G|$ disconnected parts (this follows from the fact that the projection of a connected space cannot be disconnected). However, this is impossible because the domain of $f$ is $R^N$, which is connected, and $f$ is an homeomorphism, thus preserving the connectedness of $R^N$.

In summary, we have shown that, for topological reasons, a continuous invertible encoder cannot possibly disentangle the action of a finite group acting on image space $R^N$. In the next section, we show that topological defects arise in the neighborhood of all images presenting a symmetry with respect to the transformation.

## C.2 Topological defects arise in the neighborhood of symmetric images

We proved in the previous section that it is impossible to map a finite group acting on $R^N$ to a disentangled representation with a continuous invertible encoder. In this section, in order to gain intuition of why disentanglement is impossible, we show that topological defects appear in the neighborhood of images that present a symmetry with respect to the group action.

### C.2.1 $f_E$ is not continuous about symmetric images: Formal Proof

Let's consider an image $x_s$ that presents a symmetry with respect to the group action:

$$\exists g \in G, g \neq e_G, g x_s = x_s \tag{17}$$

Let's further assume that an infinitesimal perturbation to this image along a direction $u$ breaks the symmetry of the image:

$$\forall 0 < \epsilon < E, x' := x_s + \epsilon u, g x' \neq x' \tag{18}$$

Since $f_E$ preserves the information about the transformation,

$$|f_E(gx') - f_E(x')| > C \neq 0 \tag{19}$$

where $C$ is the smallest distance between two disconnected points of $M_E$. We assume that the group action is continuous:

$$gx' = gx_s + O(\epsilon) = x_s + O(\epsilon) \tag{20}$$

We can rewrite equation 19 as:

$$|f_E(x_s + O(\epsilon)) - f_E(x_s + O(\epsilon))| > C \neq 0 \tag{21}$$

which is in contradiction with the continuity hypothesis on the encoder. We have thus shown that the equivariant part of the encoder $f_E$ presents some discontinuities around all images that present a symmetry. Note that for both rotations and translations, the uniform image is an example of symmetric image with respect to these transformations.

### C.2.2 $f_I$ IS NOT DIFFERENTIABLE ABOUT SYMMETRIC IMAGES: VISUAL PROOF

As an example, we consider an equilateral triangle which is either perturbed at its top corner, left corner, or both corners (Fig. 4). When perturbed on either one of its corner, the perturbation moves the image to the same orbit, because the triangle perturbed on its right corner is a rotated version of the triangle perturbed on its top corner. The gradient of $f_I$ along these two directions at the equilateral triangle image should thus be the same (so as not to leak information about the transformation in the invariant subspace $Z_I$). The simultaneous perturbation along the two corners moves the image to a different orbit, so the gradient of $f_I$ along this direction should not be aligned with the previous gradients (so as to preserve all information about identity of the orbit). And yet, if the function $f_I$ was differentiable everywhere, this gradient should be a linear combination of the former gradients, and thus all three gradients should be collinear. The function $f_I$ can thus not be differentiable everywhere. This imperative is incompatible with many deep learning frameworks, where the encoder is implemented by a neural network that is differentiable everywhere (with a notable exception for networks equipped with relu nonlinearities which are differentiable almost everywhere).

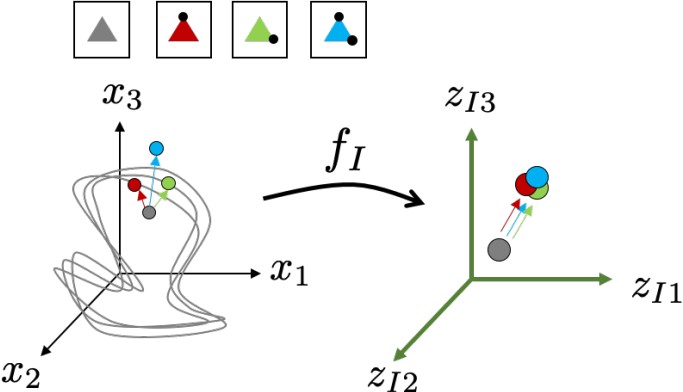

Figure 4: Visual proof that the invariant part of the encoder $f_I$ cannot be differentiable about symmetric figures. We assume $f_I$ is differentiable and show a contradiction. We consider an equilateral triangle which is perturbed at its top corner, left corner, or both corners. When perturbed either one of its corner, the perturbation brings the image to the same orbit, because of the symmetry. In latent space, the perturbation should thus move the latent representation in the same direction. The perturbation along the two corners simultaneously brings to image to a different orbit, and yet, since the perturbation is a simple linear combination of the single-corner perturbations, it can only be colinear to these perturbations. This collinearity leads to the encoder not being injective, and thus loosing information about the identity of the image.

### C.2.3 $f_I$ IS NOT DIFFERENTIABLE ABOUT SYMMETRIC IMAGES: FORMAL PROOF

Next, we show that, if we add an extra assumption on the encoder, namely that it is differentiable everywhere, it is also impossible to achieve the invariant part of the encoder $f_I$. Note that this extra assumption if true for networks equipped with differentiable non-linearities, such as tanh or sigmoid, but not for networks equipped with relus.

Let's consider an image $x_s$ presenting a symmetry w.r.t the group action. We consider perturbations of that image along two distinct directions $u$ and $gu$.

By symmetry, it is easy to see that:

$$\left.\frac{\partial f_I}{\partial u}\right|_{x_s} = \left.\frac{\partial f_I}{\partial gu}\right|_{x_s} \tag{22}$$

As a consequence, a perturbation by $u' = \frac{(u+gu)}{2}$, is equal to the perturbation along one or the other direction:

$$\left.\frac{\partial f_I}{\partial u'}\right|_{x_s} = \frac{1}{2} * \left(\left.\frac{\partial f_I}{\partial u}\right|_{x_s} + \left.\frac{\partial f_I}{\partial gu}\right|_{x_s}\right) = \left.\frac{\partial f_I}{\partial u}\right|_{x_s} \tag{23}$$

In the general case, $x' = x_s + \epsilon u'$ does not belong to the same orbit as $x = x_s + \epsilon u$. $f_I$ is thus losing information about which orbit the perturbed image $x'$ belongs to, which is in contradiction with the assumption shown in Equation 16.

### C.3 CHARACTER THEORY OF THE DISENTANGLEMENT OF FINITE DISCRETE LINEAR TRANSFORMATIONS

#### C.3.1 THE SHIFT OPERATOR

Datasets are structured by group transformations that act on the data samples. Our goal is for the model to reflect that structure. Specifically, we want our model to be equivariant to the transformations that act on the data. We defined group equivariance in Section A.

We show here that a carefully chosen distributed operator in latent space, —the shift operator— is linearly isomorphic to all cyclic linear transformations of finite order $K$ of images (i.e. the group $G$ of transformations is cyclic and finite and it acts linearly on image space). The practical consequences of this fact is that it is possible to learn an equivariant mapping to every affine transformation using this operator, using linear invertible encoder and decoder architectures.

Consider a linear encoder model $f = W$. If we want $W$ to be an equivariant invertible linear mapping $W$ between $X$ and $Z$, Equation 8 rewrites as follows:

$$W \phi_k(x) = \psi_k(W\,x)\ \forall x \in X(=\mathbb{R}^N), \forall k \in K \tag{24}$$

where $\phi_k$ and $\psi_k$ are the representations of $g_k \in G$ on the image and latent space respectively, as defined in A.

For $W$ to be equivariant, Equation 24 must be true for every image $x$. As $W$ is invertible, Equation 24 is true if and only if, $\forall k \in K$, the two representations $\psi_k$ and $\phi_k$ are isomorphic:

$$\forall k \in K, \phi_k = W^{-1}\,\psi_k\,W \tag{25}$$

We consider additional properties on $\phi$ corresponding to the assumptions (i) that $G$ is cyclic of order $K$ with generator $g_0$ and (ii) $\phi$ is isomorphic to the regular representation of $G$ (see Scott & Serre (1996)):

1. $\phi$ is cyclic, i.e. such that $\phi_k = \phi_0^k$ where $\phi_0 = \phi(g_0)$ and thus $\phi_0^K = I$.
2. The character of $\phi$ is s.t. $\chi_\phi(e) = N$ and $\chi_\phi(g_k) = 0$ for $g_k \neq e$.

The second property might seem counter-intuitive, but it just means that the transformation leaves no pixel unchanged (i.e. permutes all the pixel). In the case of rotations, this is approximately true since only the rotation origin remains in place. The character table of $\phi$, given the second property, is

$$
\begin{array}{c|c|c|c|c|c}
 & e & g_0 & g_0^2 & \cdots & g_0^{K-1} \\
\hline
\chi_\phi & N & 0 & 0 & 0 & 0
\end{array}
$$

We have just seen that if the encoder and decoder are linear and invertible, the two representations $\phi$ and $\psi$ must be isomorphic. Two representations are isomorphic if and only if they have the same character (Scott & Serre, 1996, Theorem 4, Corollary 2). We thus want to choose $\psi$ such that it preserves the character of the representation $\phi$ corresponding to the action of $G$ on the dataset of images. Importantly, we will see that our proposed operator needs to be *distributed* in the sense that it should act on the full latent code.

Let us consider the matrix of order $K = |G|$ that corresponds to a shift of elements in a $K$-dimensional vector by $k$ positions. It is the permutation corresponding to a shift of 1, exponentiated by $k$. We construct from $M_k$ the *shift operator* as a representation of the group's action on the latent space. For each $g_k \in G$ such that $g_k = g_0^k$, its corresponding shift operator is the block diagonal matrix of order $N$ composed of $\frac{N}{K}$ repetition of $M^k$.

$$M_k := \begin{bmatrix} 0 & 0 & \dots & 1 \\ 1 & 0 & \dots & 0 \\ 0 & 1 & 0 & \vdots \\ \vdots & & & \\ 0 & \dots & 1 & 0 \end{bmatrix}^k \quad (26) \qquad \psi_k := \begin{bmatrix} M^k & & & \\ & M^k & & \\ & & \dots & \\ & & & M^k \end{bmatrix} \quad (27)$$

Let us compute the character table of this representation. First, for the identity, is it trivial to see that $\chi_\psi(e) = N$. Second, for any $g_k \neq e$, we have

$$\chi_\psi(g_k) = Tr(\psi_k) = 0 \tag{28}$$

since all diagonal elements of $g_k$ will be 0. Therefore, the character table of the shift operator is the same as $\chi_\phi$:

$$\begin{array}{c|c|c|c|c|c} & e & g_0 & g_0^2 & \dots & g_0^{K-1} \\ \hline \chi_\psi & N & 0 & 0 & 0 & 0 \end{array}$$

Using this shift operator ensures that an equivariant invertible linear mapping $W$ exists between image space and the latent space equipped with the shift operator. Note that the character of the *disentangled operator* does not match the character table of $\phi$, and so *we verify once again in this linear autoencoder setting that the disentangled operator is unfit to be equivariant to affine transformations of images.*

When $K$ does not divide $N$, we use a latent code dimension that is slightly different than $N$ but divisible by $K$. This is an approximation of the correct theoretical operator and we verify that it works well in practice.

In the next Section C.3.2, we show that we can also replace this shift operator by a complex diagonal operator, which is more computationally efficient to multiply with the latent.

## C.3.2 COMPLEX DIAGONAL SHIFT OPERATOR

In order to optimise computational time, we can also consider for $M_k$ the following diagonal complex matrix:

$$M_k := \begin{bmatrix} 1 & 0 & \ddots & & 0 \\ 0 & \omega & & & \\ \vdots & & \ddots & & \\ 0 & & & & \omega^{K-1} \end{bmatrix}^k \tag{29}$$

with $\omega = e^{\frac{2i\pi}{K}}$. The shift operator in this case is a diagonal matrix, as follows:

$$\psi_{k,N} := \begin{bmatrix} \overbrace{\begin{matrix} 1 & & \\ & \ddots & \\ & & \omega^{K-1} \end{matrix}}^{\mathscr{M}_k} & & & & \\ & \overbrace{\begin{matrix} 1 & & \\ & \ddots & \\ & & \omega^{K-1} \end{matrix}}^{\mathscr{M}_k} & & \\ & & \ddots & \\ & & & \omega^{K-1} \end{bmatrix} \tag{30}$$

Let us compute the character table of this representation. First, for the identity, is it trivial to see that $\chi_\psi(e) = N$. Second, for any $g_k \neq e$, we have

$$\chi_\psi(g_k) = Tr(\psi_k) = \sum_{n=0}^{N-1} (\omega^k)^n = \frac{1 - (\omega^k)^N}{1 - \omega^k} = 0 \tag{31}$$

as $\omega^{kN} = e^{\frac{2i\pi kN}{K}} = 1$ since we're assuming $N$ can be divided by $K$. Again, the character table of the shift operator is the same as $\chi_\phi$. Using this operator fastens computation since it requires only multiplying the diagonal values (a vector of size $N$) with the latent code (a vector of size $N$ as well) instead of doing a matrix multiplication. Note that when using this complex version of the shift operator, the encoding and decoding layers of the autoencoder should be complex as well.

When $K$ does not divide $N$, we still use a latent code of size $N$ and the operator $\psi_k$ is a diagonal matrix of order $N$, but the last cycle $1, \omega, \dots$ is unfinished (does not go until $\omega^{K-1}$). The character of this representation is no longer equal to $0$ for non-identity elements $g_k \neq e$, but equals a small value $<< N$ and this approximation works well in practice.

## D   THE CASE OF MULTIPLE TRANSFORMATIONS: FORMAL DERIVATIONS

### D.1   TRANSLATIONS IN BOTH AXES AS A DIRECT PRODUCT

To cover the case of 2D translations (acting on both $x$ and $y$ axes of the image), we consider an abelian group $G$ that is the direct product of two subgroups $G = A_x \times A_y$. Both $A_x$ and $A_y$ are normal in $G$ because every subgroup of an abelian group is normal. Moreover, we consider that $A_x$ and $A_y$ are both cyclic of order $K$ and $K'$ respectively, which is the case for integer translation of an image using periodic boundary condition. We denote $a_{x,0}$ and $a_{y,0}$ the generators of $A_x$ and $A_y$, and write each translation as $a = (a_{x,k}, a_{y,k'})$ with $a_{x,k} = a_{x,0}^k$, $a_{y,k'} = a_{y,0}^{k'}$.

We show that the shift operator can handle this case, with differences. We assume for simplicity that both $K$ and $K'$ divide $N$, and $KK'$ divides $N$. Following Scott & Serre (1996), we write group elements $g \in G$ as $g_{(k,k')} = (a_{x,k}, a_{y,k'})$. The order of the group is $K = KK'$. If we consider the regular representation of 2D translations over the image space, as in Section C.3, its character table is

$$\begin{array}{c|c|c|c|c|c} & e & g_{(0,1)} & g_{(0,2)} & \cdots & g_{(K-1,K'-1)} \\ \hline \chi_\phi & N & 0 & 0 & 0 & 0 \end{array}$$

We consider two representations over the latent space: $\psi_x : A_x \to GL(\mathbb{C}^K)$ is a linear representation of $A_x$ and $\psi_y : A_y \to GL(\mathbb{C}^{K'})$ a linear representation of $A_y$, each of the matrix form described in Section C.3.2. Group theory (Scott & Serre, 1996) tells us that $\psi$ is the tensor product of $\psi_x$ and $\psi_y$, i.e. for $g = (k, k') \in G$:

$$\psi_k = \phi(g) = \phi((a_{x,k}, a_{y,k'})) = (\psi_x \otimes \psi_y)(a_{x,k}, a_{y,k'}) = \psi_x(a_{x,k}) \otimes \psi_y(a_{y,k'}) \tag{32}$$

Let us consider the two shift operators:

$$\psi_{x,k} := Diag(1, \omega_1, \omega_1^2 \dots, \omega_1^{K-1})^k \tag{33}$$

$$\psi_{y,k'} := Diag(1, \omega_2, \omega_2^2, \dots \omega_2^{K'-1})^{k'} \tag{34}$$

where $\omega_1 = e^{\frac{2i\pi}{K}}$ and $\omega_2 = e^{\frac{2i\pi}{K'}}$. Then the tensor product $\psi_{x,k} \otimes \psi_{y,k'}$ writes as a diagonal matrix of order $KK'$:

$$\psi_{x,k}\otimes\psi_{y,k'} := \begin{bmatrix} 1 & & & & & & & \\ & \omega_2^{k'} & & & & & & \\ & & \omega_2^{2k'} & & & & & \\ & & & \ddots & & & & \\ & & & & \omega_2^{k'(K'-1)} & & & \\ & & & & & \omega_1^{k} & & \\ & & & & & & \omega_1^{k}\omega_2^{k'} & \\ & & & & & & & \ddots \\ & & & & & & & & \omega_1^{k(K-1)}\omega_2^{k'(K'-1)} \end{bmatrix} \tag{35}$$

The character of $\psi_{x,k} \otimes \psi_{y,k'}$ is

$$\chi_{\psi_{x,k}\otimes\psi_{y,k'}} = \sum_{n=0}^{K'-1} \omega_2^{nk'} \times \sum_{n=0}^{K-1} \omega_1^{nk} \tag{36}$$

If $k \neq 0$, $\sum_{n=0}^{K-1} \omega_1^{kn} = \frac{1-\omega_1^{kK}}{1-\omega_1^{k}} = 0$ since $\omega_1^{kK} = e^{2i\pi k} = 1$ (and similarly for $\omega_2$ if $k' \neq 0$). Hence, for $(k,k') \neq (0,0)$, $\sum_{n=0}^{K'-1} \omega_2^{nk'} \times \sum_{n=0}^{K-1} \omega_1^{nk} = 0$. For $(k,k') = (0,0), \chi_{\psi_{x,k}\otimes\psi_{y,k'}} = KK'$. Thus, the character table of $\psi_x \otimes \psi_y$ is

| | e | $g_{(0,1)}$ | $g_{(0,2)}$ | $\cdots$ | $g_{(K-1,K'-1)}$ |
|---|---|---|---|---|---|
| $\chi_\psi$ | $KK'$ | 0 | 0 | 0 | 0 |

We will use for 2D translations a diagonal operator that is the repetition of $\frac{N}{KK'}$ times $\psi_{x,k} \otimes \psi_{y,k'}$ (assuming $KK'$ divides $N$), denoted $\psi$, with is for $g_{k,k'} = (a_{x,k}, a_{y,k'})$

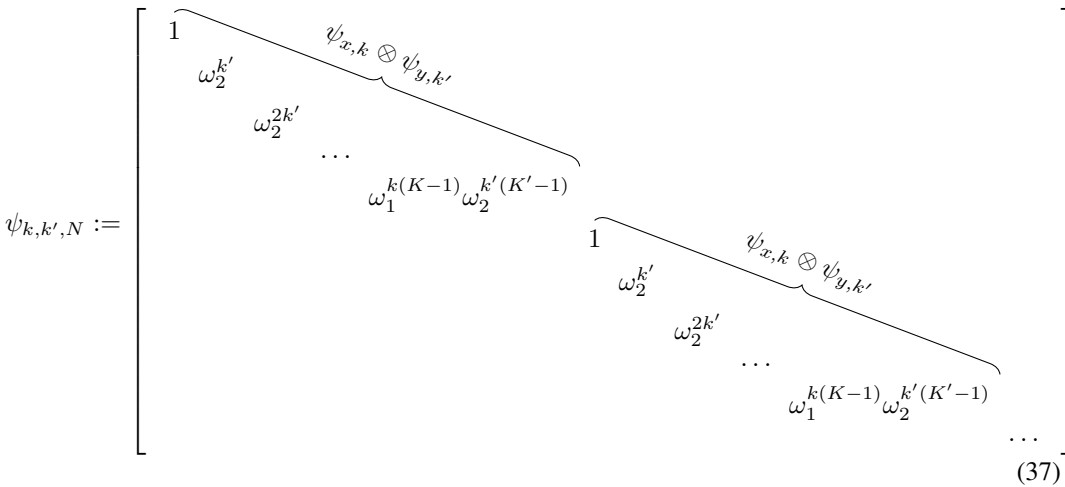

$$\tag{37}$$

Thus, the character table of $\psi$ is

| | e | $g_{(0,1)}$ | $g_{(0,2)}$ | $\cdots$ | $g_{(K-1,K'-1)}$ |
|---|---|---|---|---|---|
| $\chi_\psi$ | $N$ | 0 | 0 | 0 | 0 |

We see that $\psi$ has the same character table as $\psi$ the representation in image space, hence is a suited operator to use for this case.

### D.2 INSIGHTS ON THE STACKED SHIFT OPERATORS MODEL FOR THE 2D TRANSLATION GROUP

For the case of 2D translations where $G = A_x \times A_y$, we first use the operator corresponding to $a_{x,k}$, the translation in $x$, then intersect a linear layer denoted $L_1$ before using the operator corresponding

to $a_{y,k'}$, the translation in $y$. When we operate on the latent code, we perform

$$z' = \psi_{y,k',N} L_1 \psi_{x,k,N} z \tag{38}$$

where $\psi_{x,k,N}$ is the operator representing the translation in $x$. It is a matrix of order $N$, where $\psi_{x,k}$ is repeated $\frac{N}{K}$ times.

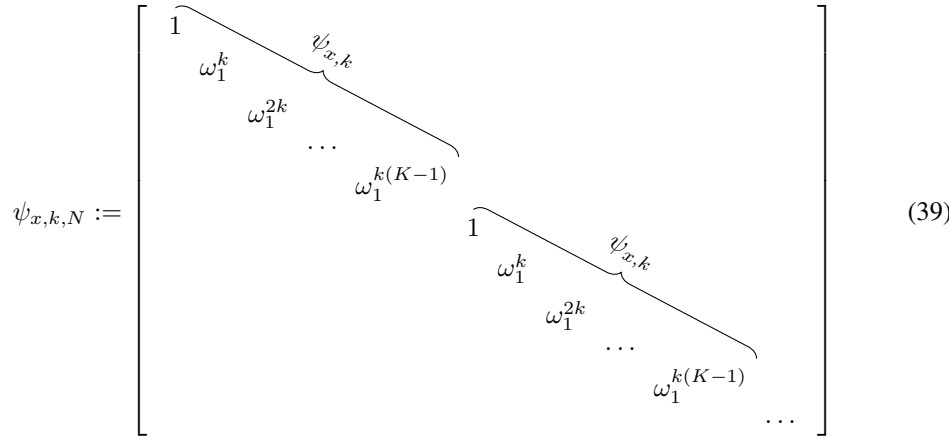

$$\tag{39}$$

Similarly, the operator corresponding to $a_{y,k'}$ is a matrix of order $N$, where $\psi_{y,k'}$ is repeated $\frac{N}{K'}$ times

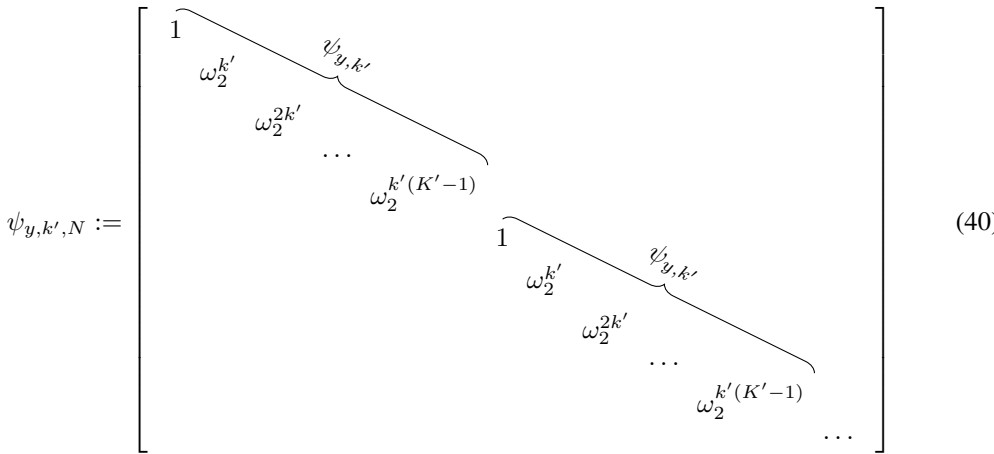

$$\tag{40}$$

We see that in order for the product $\psi_{y,k',N} L_1 \psi_{x,k,N}$ when applied to $z$, to match the result of $\psi_{k,k',N}$ applied to $z$, we need a permutation matrix $P$ that operates on a matrix of order $KK'$ made of blocks of $K'$ times $\psi_{x,k}$ and returns a matrix of order $KK'$ with:

- 1 at the first $K'$ rows and columns $c = 1, (K+1), \ldots KK' - K + 1$
- $\omega_1^k$ at rows from $K' + 1$ to $2K$ and columns $c = 2, K + 2, \ldots KK' - K + 2$ etc.
- until $\omega_1^{k(K-1)}$ at rows from $KK' - K'$ to $KK'$ and columns $c = K, 2K, KK'$

And the layer $L_1$ would be a matrix of order $N$ made of this permutation matrix $P$, repeated in block diagonal form $\frac{N}{KK'}$ times, such that $(\psi_{y,k',N} L_1 \psi_{x,k,N}) z = \psi_{k,k',N} z$.

### D.3 SEMI-DIRECT PRODUCT OF TWO GROUPS

#### D.3.1 DEFINITION OF A SEMI-DIRECT PRODUCT

A semi-direct product $G = A \rtimes H$ of two groups $A$ and $H$ is a group such that:

- $A$ is normal in $G$.

- There is an homomorphism $f : H \to Aut(A)$ where $Aut(A)$ is the group of automorphism of $A$. For $a \in A$, denote $f(h)a$ by $h(a)$. In other words, $f(h)$ represents how $H$ acts on $A$.
- The semi-direct product $G = A \rtimes H$ is defined to be the product $A \times H$ with multiplication law

$$(a_1, h_1)(a_2, h_2) = (a_1 h_1(a_2), h_1 h_2) \tag{41}$$

Note that this enforces that $h_1(a_1) = h_1 a_1 h_1^{-1}$.

### D.3.2 IRREDUCIBLE REPRESENTATION OF A SEMI-DIRECT PRODUCT

In this section, we also assume $A$ is abelian. One can derive the irreducible representations of $G$ in this case, as explained in Scott & Serre (1996) Section 8.2.

First, consider the irreducible characters of $A$, they are of degree 1 since $A$ is abelian and form the character group $X = Hom(A, \mathbb{C}^*)$. We use $X$ to match Scott & Serre (1996) notation, but note that $X$ does not denote image space here, but the group of characters of $A$. $H$ acts on this group by:

$$h_\chi(a) = \chi(h^{-1}ah) \tag{42}$$

Second, consider a system of representative of the orbits of $H$ in $X$. Each element of this system is a $\chi_i$, $i \in G/H$. For a given $\chi_i$, denote $H_i$ the stabilizer of $\chi_i$, i.e. $h_{\chi_i} = \chi_i$. This means

$$h_{\chi_i}(a) = \chi_i(h^{-1}ah) = \chi_i(a), \forall a \in A \tag{43}$$

Third, extend the representations of $A$ to a representation of $G_i = A \rtimes H_i$ by setting

$$\chi_i(ah) = \chi_i(a), h \in H_i, a \in A \tag{44}$$

The $\chi_i$ are also characters of degree 1 of $G_i$.

Fourth, consider the irreducible representations of $H_i$. Scott & Serre (1996) propose to use the irreducible representation $\rho$ of $H_i$. Combining with the canonical projection $G_i \to H_i$ we get irreducible representations $\tilde{\rho}$ of $G_i$. Irreducible representations of $G_i$ are obtained by taking the tensor product $\chi_i \otimes \tilde{\rho}$.

Finally, the irreducible representations of $G$ are computed by taking the representation induced by $\chi_i \otimes \tilde{\rho}$.

Etingof et al. (2009) show that the character of the induced representation is

$$\chi_{Ind^G_{G_i}(\chi_r \otimes \rho)}(a, h) = \frac{1}{|H_i|} \sum_{h' \in H \, s.t. \, h'^{-1}hh' \in H_i} \chi_r(h(a))\chi_\rho(h'^{-1}hh') \tag{45}$$

### D.4 REPRESENTATIONS OF THE (DISCRETE FINITE) SPECIAL EUCLIDEAN GROUP

In this section, we focus on the specific case of the semi-direct product $G = A \rtimes H$ where $A = A_x \times A_y$ is the group of 2D translations and $H$ the group of rotations. Hence, $A$ is abelian and $H$ is a cyclic group. We will derive the irreducible representations of this group using the method presented in the previous section D.3.2.

### D.4.1 A NOTE ON THE DISCRETE FINITE SPECIAL EUCLIDEAN GROUP

While the Special Euclidean group respects the structure of a semi-direct product in the continuous transformations case, we will consider its discrete and finite version. That is, we consider a finite number of translations and rotations. With integer valued translation, which is of interest when working with images (considering translations of a finite number of pixels), we cannot consider all rotations. Indeed for example rotations of $2\frac{\pi}{8}$ break the normal aspect of the subgroup of translations.

*Proof.* Take the translation element $a = (1, 1)$ (one pixel translation in $x$ and $y$ respectively), and the rotation $h$ of angle $2\frac{\pi}{8}$. Consider the composition $hah^{-1}$, applied to a point of coordinates $i, j$ this gives $hah^{-1}(i, j) = h(h^{-1}(i, j) + (1, 1)) = (i, j) + h(1, 1) = (i, j) + (0, \sqrt{2})$ and the translation $(0, \sqrt{2})$ is not an integer translation. Thus, $A$ is not normal in $G$ in this case. $\qquad \square$

In what follows, we consider rotations that preserve the normality of the group of integer 2D translations of the image. Namely, these are rotations of $\frac{2\pi}{4}$, $\pi$, $\frac{3\pi}{4}$ and identity and their multiples. Nonetheless, we think the approach is insightful and approximate solutions could be found with this method. Furthermore, to ease derivations we consider that both $K \geq 2$ and $K' \geq 2$ are odd (and thus the product $KK'$ is odd), such that stabilizers of the character group of $A$ are either the entire $H$ or only the identity. We leave the exploration of even $K$ and $K'$ for future work.

### D.4.2 FINDING THE ORBITS

As we consider integer translations with periodic boundary conditions, characters of $A$ are 2D complex numbers and are function of $\mathbb{Z}^2$ elements of the 2D discrete and finite translation group of the form $a = (x, y)$ with $x = x_0^k$ and $y = y_0^{k'}$. The characters of this group are evaluated as $\chi_{(x_1, y_1)}(a) = e^{i2\pi(\frac{x_1}{K}k + \frac{y_1}{K}k')}$, where $x_1, y_1$ are of the form with $x_1 \in 1 \ldots K$ and $y_1 \in 1 \ldots K'$, respectively.

We consider two cases:

1. $\chi_{0,0}$.

2. $\chi_{(x_1, y_1)}$ with $x_1 \neq 0$ or $y_1 \neq 0$.

Note that the total number of orbits of $H$ in $X$ is $\frac{1}{H} \sum_{\chi_x \in X} |H_x|$ where $H_x$ is the stabilizer of $\chi_x$. Either $\chi_x = e_X$ is the identity element of $X$ and $|H_x| = |H|$ or $\chi_x \neq e_X$ and the only stabilizer is $e_H$ (as shown in Section D.6.1) thus $|H_x| = 1$. Thus, we have that:

$$\frac{1}{H} \sum_{\chi_x \ inX} |H_x| = \frac{1}{H}(1 \times |H| + (|A| - 1) \times 1) = 1 + \frac{|A| - 1}{|H|} \tag{46}$$

Hence, there are $\frac{|A|-1}{|H|}$ orbits in case 2 (the total number of orbits minus the orbit considered in the first case, $\chi_{(0,0)}$).

### D.4.3 ACTION OF $H$ ON THE CHARACTERS OF $A$

Elements of $H$ acts on characters of $A$ as:

$$h_{\chi_{(x_1,y_1)}}(a) = \chi_{(x_1,y_1)}(h^{-1}ah) \tag{47}$$

$$= \chi_{(x_1,y_1)}(h^{-1}(a)) \tag{48}$$

$$= e^{i2\pi(\frac{x_1}{K}(\cos(-\theta)x - \sin(-\theta)y) + \frac{y_1}{K'}(\sin(-\theta)x + \cos(-\theta)y))} \tag{49}$$

$$= e^{i2\pi((\frac{x_1}{K}\cos(\theta) + \frac{y_1}{K'}\sin(-\theta))x + (\frac{x_1}{K}(-\sin(-\theta)) + \frac{y_1}{K'}\cos(-\theta))y)} \tag{50}$$

$$= e^{i2\pi((\frac{x_1}{K}\cos(\theta) - \frac{y_1}{K'}\sin(\theta))x + (\frac{x_1}{K}\sin(\theta) + \frac{y_1}{K'}\cos(\theta))y)} \tag{51}$$

$$= \chi_{(x_1 \cos(\theta) - y_1 \sin(\theta), x_1 \sin(\theta) + y_1 \cos(\theta))}(a) \tag{52}$$

$$= \chi_{h((x_1,y_1))}(a) \tag{53}$$

where $\theta$ is the angle of rotation of $h$.

### D.4.4 CASE 1 $\chi_{0,0}$ (ORBIT OF THE ORIGIN)

A representative is $\chi_{0,0} = 1$. The stabilizer group $H_i$ is the entire $H$, and the irreducible representations of $H$ are of the form $e^{i2\pi \frac{n}{|H|}}$ for $n \in 1, \ldots, |H|$ where $|H|$ is the total number of rotations. Thus we use the tensor product $\theta_n(0, \phi) = 1 \otimes e^{i2\pi \frac{n}{|H|}}$ as a representation of $G$. There are $|H|$ of such irreducible representations of $G$, all of degree 1, since the group of rotation is abelian.

The resulting representations, corresponding to the irreducible we get from combining this

orbit and each one of the irreducible of $H$ can be represented in matrix form:

$$\rho(a,h) := \begin{bmatrix} 1 & 0 & \ddots & & 0 \\ 0 & e^{i2\pi\frac{1}{|H|}} & & & \\ \vdots & & \ddots & & \\ 0 & & & & e^{i2\pi\frac{|H|-1}{|H|}} \end{bmatrix}^{j} \tag{54}$$

where $j$ is such that $h = h_0^j$ and $h_0$ is the generator of the group of rotations.

### D.4.5    CASE 2 $\chi_{(x_1,y_1)}$ ($\frac{|A|-1}{|H|}$ OF THEM)

A representative can be taken to be $\chi_r = \chi_{(x_1,y_1)}(x,y)$, and $x_1, y_1$ are now fixed. Its stabilizer group $H_i$ is only $\{e_H\}$ (see (Berndt, 2007) and proof in Section D.6.1).

We now select $\rho$ an irreducible representation of $\{e_H\}$, and we select the trivial representation 1 and combining with the canonical projection step 1 will also be a representation of $G_i$. We then take the tensor product $\chi_r \otimes 1$ as a representation of $G_i = A \rtimes \{e_H\}$. Let us now derive the representation of the entire group $A \rtimes G = Ind_{G_i}^G(\chi_r \otimes 1)$.

**Induced representation** $Ind_{G_i}^G(\chi_r \otimes 1)$    First, we need a set of representatives of the left coset of $G$ in $G/G_k = A \rtimes e_H$. The left cosets are defined as $(a,h)G_k = \{(a,h)(a_k,e_H) \, \forall a_k \in A\}$. There are $|H|$ cosets (one per $h$), which we can denote $(e_A,h)G_k$. We take a representative $g_i$ for each coset $G_k$. We take as representative $(e_A,h_i) = (e_A,h_i)(e_A,e_H) \in (e_A,h)G_k$ [1], so the $h_i$ are now fixed.

The representation $(\rho, W)$ is induced by $(\chi_r \otimes 1, V)$ if

$$W = \bigoplus_{i=1}^{|H|} \rho(g_i)V \tag{55}$$

with $g_i = (e_A, h_i)$ and $\rho(g_i)$ is described below.

For each $g$ and each $g_i$, there is $j(i) \in \{1, \ldots |H|\}$ and $f_j \in G_k$ such that we have $gg_i = g_{j(i)}f_j$. Indeed:

$$gg_i = (a,h)(e_A,h_i) \tag{56}$$
$$= (a, hh_i) \tag{57}$$
$$= (e_A, hh_i)((hh_i)^{-1}(a), e_H) \tag{58}$$

so $g_{j(i)} = (e_A, hh_i)$. In other words, the action of $g = (a,h)$ on an element $w \in W$ permutes the representatives:

$$\rho(g)w = \rho((a,h))\sum_{i=1}^{|H|}\rho(g_i)v_i = \sum_{i=1}^{|H|}\rho(g_{j(i)})\chi((hh_i)^{-1}(a))v_i \tag{59}$$

Now, we need to find for each element of $G$, the resulting permutation of the coset representatives. $G$ is generated by $a \in A, h \in H$: any element $(a,h) \in G$ can be written as $(a,e_H)(e_A,h)$. For $(a, e_H)$, we get $g_{j(i)} = (e_A, h_i) = g_i$. The induced representations are thus:

$$\rho(a, e_H) = \begin{bmatrix} \chi_r(h_1^{-1}(a)) & \ldots & 0 & 0 \\ 0 & \chi_r(h_2^{-1}(a)) & \ldots & 0 \\ 0 & & \ldots & 0 \quad \chi_r(h_{|H|}^{-1}(a)) \end{bmatrix} \tag{60}$$

---

[1]An element $(a,h)$ is in the same coset as $(e_A,h)$ as $(a,h)^{-1}(e_A,h) = (h^{-1}(a^{-1}), h^{-1})(e_A, h) = (h^{-1}(a^{-1}), e_H) \in G_k$

with no permutation. For $(e_A, h)$ we get $g_{j(i)} = (e_A, hh_i)$. The resulting induced representations are:

$$\rho(e_A, h) = \begin{bmatrix} 0 & 0 & \cdots & \chi_r((hh_{j^{-1}(1)})^{-1}(e_A)) \\ 0 & \chi_r((hh_{j^{-1}(2)})^{-1}(e_A)) & \cdots & \\ 0 & \cdots & \chi_r((hh_{j^{-1}(|H|)})^{-1}(e_A)) & 0 \end{bmatrix} \tag{61}$$

i.e.

$$\rho(e_A, h) = P_h = \begin{bmatrix} 0 & 0 & \cdots & 1 \\ 0 & 1 & \cdots & \\ 0 & \cdots & 1 & 0 \end{bmatrix} \tag{62}$$

Where the permutation matrix $P_h$ above represents how $h$ acts on its own group. If $hh_i = h_j$ then $P_h$ has a 1 at the column $j$ of its $i$-th row. For example, if $hh_{|H|} = h_1$ then $j(|H|) = 1$, $j^{-1}(1) = |H|$ and so for the row 1, there is $\chi_r((hh_H)^{-1}(e_A)) = 1$ at the $|H|$-th column as above.

Let us denote for clarity $\chi_{r,i} = \chi_r(h_i^{-1}(a))$. The representations of $(a, h)$ will be

$$\rho(a, h) = \rho(a, e_H)\rho(e_A, h) = \begin{bmatrix} \chi_{r,1} & \cdots & 0 & 0 \\ 0 & \chi_{r,2} & \cdots & 0 \\ 0 & \cdots & 0 & \chi_{r,|H|} \end{bmatrix} P_h = \begin{bmatrix} 0 & 0 & \cdots & \chi_{r,1} \\ 0 & \chi_{r,2} & \cdots & \\ 0 & \cdots & \chi_{r,|H|} & 0 \end{bmatrix} \tag{63}$$

For each orbit, we get one of these (of degree $H$).

### D.4.6 RESULTING REPRESENTATION

The resulting representation will be a block diagonal matrix of size $|A||H|$ of the form

$$\tag{64}$$

The first $|H|$ elements on the diagonal correspond to the irreducible representation of $h = h_0^j$, this is the shift operator we have been considering in the single transformation case. Second, each matrix product $M_r(a, h)P_h$, representative $r$, is repeated $|H|$ times as it is of degree $|H|$ (see calculation of degree in the Section D.6.2) and corresponds to:

$$M_r(a) = \begin{bmatrix} \chi_r(h_1^{-1}(a)) & \cdots & 0 & 0 \\ 0 & \chi_r(h_2^{-1}(a)) & \cdots & 0 \\ 0 & \cdots & 0 & \chi_r(h_{|H|}^{-1}(a)) \end{bmatrix} \tag{65}$$

If we assume $|A||H|$ divides $N$, we can use an operator $\psi_{a,h,N}$ that is the repetition of $\frac{N}{|A||H|}$ times $\rho(a,h)$ into a matrix of order $N$:

$$
\psi_{a,h,N} := \begin{bmatrix}
1 & & \overbrace{\qquad\qquad}^{\rho(a,h)} & & & & \\
e^{i2\pi\frac{1}{|H|}j} & & & & & & \\
 & \cdots & & & & & \\
 & & M_R(a)P_h & & & & \\
 & & & 1 & & \overbrace{\qquad}^{\rho(a,h)} & \\
 & & & e^{i2\pi\frac{1}{|H|}j} & & & \\
 & & & & \cdots & & \\
 & & & & & M_R(a)P_h & \\
 & & & & & & \cdots
\end{bmatrix} \tag{66}
$$

Its character table is:

| | $(e_A, e_H)$ | $(a_1, e_H)$ | $(a_2, e_H)$ | ... | $(a_{|A|}, h_{|H|})$ |
|---|---|---|---|---|---|
| $\chi_\psi$ | $N$ | $0$ | $0$ | $0$ | $0$ |

and is the same character as the operator acting on image space, as shown with the computation of the character of $\rho(a,h)$ in Section D.4.7.

### D.4.7 CHARACTER TABLE

For the identity element $(e_A, e_H)$, we get for $\rho(a,h)$ the identity matrix, and its trace is $|A||H|$. When $h \neq e_H$, $P_h$ has no diagonal element and $\sum_{k=0}^{|H|-1} e^{i2\pi\frac{k}{|H|}k'} = 0$, hence the trace of $\rho(a,h) = 0$. When $a \neq e_A$, if $h \neq e_H$ we are in the previous case. If $h = e_H$, $P_H$ is the identity matrix and the representation will be

$$
\rho(a, e_H) = \begin{bmatrix}
1 & & & & & & & \\
 & 1 & & & & & & \\
 & & \cdots & & & & & \\
 & & & 1 & & & & \\
 & & & & M_{r_1}(a)I & & & \\
 & & & & & M_{r_1}(a)I & & \\
 & & & & & & \cdots & \\
 & & & & & & & M_R(a)I
\end{bmatrix} \tag{67}
$$

The trace of $\rho(a, e_H)$ is

$$
Tr(\rho(a, e_H)) = |H| + |H| \sum_{h_i \in H} \sum_{r=r_1}^{R} \chi_r(h_i^{-1}(a)) \tag{68}
$$

as each block-diagonal matrix is repeated $|H|$ times. If we interchange the order of summation we get:

$$
Tr(\rho(a, e_H)) = |H| + |H| \sum_{r=r_1}^{R} \sum_{h_i \in H} \chi_r(h_i^{-1}(a))) \tag{69}
$$

$$
= |H| + |H| \sum_{r=r_1}^{R} \sum_{h_i \in H} h_{i,\chi_r}(a)) \tag{70}
$$

where $h_{i,\chi_r}$ represents the action of $h_i$ on $\chi_r$. The action of every $h_i$ on $\chi_r$ gives all the elements in the orbit of $\chi_r$, so the double sum results in the sum over all characters $\chi \in X$ of $A$, apart from the orbit of $\chi_{0,0}$ which contains only $\chi_{0,0}$.

$$
Tr(\rho(a, e_H)) = |H| + |H| \sum_{\chi \in X, \chi \neq X_{0,0}} \chi(a) \tag{71}
$$

$$
\tag{72}
$$

The sum of the all the irreducible characters, for $a \neq e_A$ is 0 (Scott & Serre, 1996, Corollary 2): $\sum_{\chi \in X} \chi(a) = 0$. Hence,

$$\sum_{\chi \in X} \chi(a) = 0 = X_{0,0}(a) + \sum_{\chi \in X, \chi \neq X_{0,0}} \chi(a) \tag{73}$$

$$= 1 + \sum_{\chi \in X, \chi \neq X_{0,0}} \chi(a) \tag{74}$$

$$\rightarrow \sum_{\chi \in X, \chi \neq X_{0,0}} \chi(a) = -1. \tag{75}$$

Hence,

$$Tr(\rho(a, e_H)) = |H| + |H|(-1) = 0. \tag{76}$$

Consequently, the character table of $\rho$ is the following:

| $\chi_\rho$ | $(e_A, e_H)$ | $(a_1, e_H)$ | $(a_2, e_H)$ | ... | $(a_{|A|}, h_{|H|})$ |
|---|---|---|---|---|---|
| | $|A||H|$ | $0$ | $0$ | $0$ | $0$ |

The character of $\psi$ is $\frac{N}{|A||H|}$ times the character of $\chi_\rho$, i.e.

| $\chi_\psi$ | $(e_A, e_H)$ | $(a_1, e_H)$ | $(a_2, e_H)$ | ... | $(a_{|A|}, h_{|H|})$ |
|---|---|---|---|---|---|
| | $N$ | $0$ | $0$ | $0$ | $0$ |

## D.5 INSIGHTS ON THE STACKED SHIFT OPERATORS MODEL FOR THE DISCRETE FINITE SPECIAL EUCLIDEAN GROUP

Similarly to 2D translation case, we can use the theoretical form of the representation to use to gain insight on what the intersected layers $L_i$ should be for the case of 2D translations in conjunction with rotations. Elements of this group are $(a, h)$ where $a$ is the 2D translation, composed of $a_{x,k}$ the translation on the $x$-axis, $a_{y,k'}$ translation on the $y$-axis, and $h$ the rotation. Using the stacked shift operators model, we first use the operator corresponding to the rotation, then the one for translation in $x$, then the one for $y$-translation. The resulting transformed latent code is

$$z' = \psi_{y,k',N} L_2 \psi_{x,k,N} L_1 \psi_{h,j,N} z \tag{77}$$

with $a_{y,k'} = a_{y,0}^{k'}$, $a_{x,k} = a_{x,0}^k$, $h = h_0^j$. The operators for translations $\psi_{x,k,N}$, $\psi_{y,k',N}$ are described in Equations 39 and 40. We repeat them here for clarity.

$$\psi_{a_{x,k},k,N} := \begin{bmatrix} 1 & & & & & & & & \\ & \omega_1^k & & & & & & & \\ & & \omega_1^{2k} & & & & & & \\ & & & \ddots & & & & & \\ & & & & \omega_1^{k(K-1)} & & & & \\ & & & & & 1 & & & \\ & & & & & & \omega_1^k & & \\ & & & & & & & \omega_1^{2k} & \\ & & & & & & & & \ddots \\ & & & & & & & & & \omega_1^{k(K-1)} \\ & & & & & & & & & & \ddots \end{bmatrix} \tag{78}$$

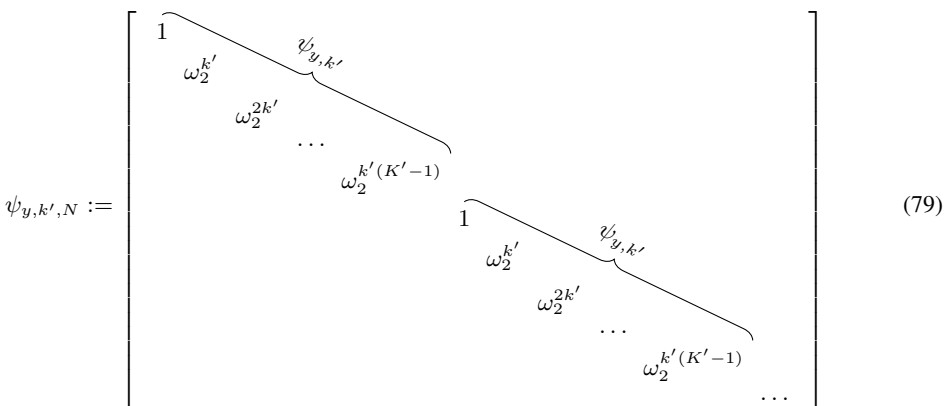

$$\psi_{y,k',N} := \quad (79)$$

where $\omega_1 = e^{\frac{2i\pi}{K}}$ and $\omega_2 = e^{\frac{2i\pi}{K'}}$. And $\psi_{h,j,N}$ is the repetition of the shift operator for the rotation $\frac{N}{|H|}$ times, as follows,

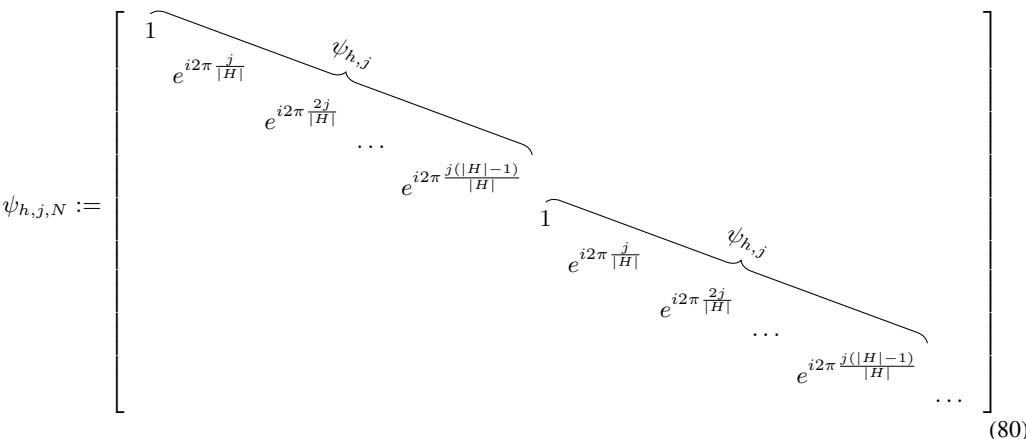

$$\psi_{h,j,N} := \quad (80)$$

The representations for each transformation $\psi_{x,k,N}$, $\psi_{y,k',N}$, $\psi_{h,j,N}$ are linked with the representation that the theory gives us in Equation 66.

First, recall that elements of $H$ acts on a character $\chi_r = \chi_{(x_1,y_1)}$:

$$\chi_{(x_1,y_1)}(h_i^{-1}(a)) = \chi_{h_i((x_1,y_1))}(a) \quad (81)$$

Thus when $h_i$ spans $H$ in a given matrix $M_r$, we obtain $|H|$ distinct characters built from $\chi_r$, i.e.

$$M_r(a) = \begin{bmatrix} \chi_{h_1(r)}(a)) & ... & 0 & 0 \\ 0 & \chi_{h_2(r)}(a) & ... & 0 \\ 0 & ... & 0 & \chi_{h_{|H|}(r)}(a) \end{bmatrix} \quad (82)$$

For another matrix $M_{r'}$, we obtain again $|H|$ distinct orbits, distinct from the one present in $M_r$ (otherwise they would be in the same orbit). We have $\frac{A-1}{H}$ representatives (excluding $\chi_{(0,0)}$), thus $|A| - 1$ characters of $A$ evaluated at $a$ are obtained by considering all the matrices $M_r$. The remaining character is $\chi_{(0,0)} = 1$. Hence, the diagonal matrix $\rho(a, e_H)$ of order $|A||H|$ contains all the characters of $A$ evaluated at the element $a$, each one repeated $|H|$ times since each matrix $M_r$ is repeated $|H|$ times, and the first diagonal elements of $\rho(a, e_H)$ are 1.

So we see that what we need are $|H|$ times each character of $A$. As explained in Section D.4.2, these characters are of the form:

$$\chi_{(x_1,y_1)}(a) = e^{i(2\pi\frac{x_1}{K}k + 2\pi\frac{y_1}{K'}k')} = e^{i(2\pi\frac{x_1}{K}k)}e^{i(2\pi\frac{y_1}{K'}k')} = \omega_1^{x_1 k}\omega_2^{y_1 k'}. \quad (83)$$

The characters of each translation group are $\chi_{x_1} = \omega_1^{x_1}$ for the $x$-translation and $\chi_{y_1} = \omega_2^{y_1}$ for the $y$-translation, thus:

$$\chi_{(x_1,y_1)}(a) = \chi_{(x_1,y_1)}(a_{x,k}, a_{y,k'}) = \omega_1^{x_1 k} \omega_2^{y_1 k'} = \chi_{x_1}(a_{x,k})\chi_{y_1}(a_{y,k'}) \tag{84}$$

Since $\psi_{x,k}$ is represented $\frac{N}{K}$ times in $\psi_{x,k,N}$ and $\psi_{y,k'}$ is represented $\frac{N}{K'}K'$ times in $\psi_{y,k',N}$, we can take $L_2$ to reorder $\psi_{y,k',N}$ into blocks of $K'$ repeated diagonal elements such that when multiplied with $\psi_{x,k,N}$ by doing $\psi_{y,k',N}L_2\psi_{x,k,N}$, we obtain $\frac{N}{KK'} = \frac{N}{|A|}$ blocks of size $|A|$, each containing every character of $A$ once. Furthermore, we can see that the resulting matrix $\psi_{y,k',N}L_2\psi_{x,k,N}$ is composed of $\frac{N}{|A||H|}$ blocks of size $|A||H|$ containing every character of $A$, but repeated $|H|$ times.

This resulting matrix is then multiplied with the matrix $L_1\psi_{h,j,N}$. So let us now turn to the representation of the rotation $h$, that is $\psi_{h,j,N}$. It is composed of the elements that appear in the upper-left diagonal of $\rho(a,h)$. But we also need to make the permutation matrices $P_h$ appear. Recall that diagonal matrix of order $|H|$ that corresponds to the shift representation of $h = h_0^j$ (not repeated $\frac{N}{|H|}$ times) is

$$\psi_{h,j} = \begin{bmatrix} 1 & & & & \\ & e^{i2\pi\frac{j}{|H|}} & & & \\ & & e^{i2\pi\frac{2j}{|H|}} & & \\ & & & \ddots & \\ & & & & e^{i2\pi\frac{j(|H|-1)}{|H|}} \end{bmatrix} \tag{85}$$

If we right-multiply $\psi_{h,j}$ by

$$C = \begin{bmatrix} 1 & 1 & \cdots & 1 \\ 1 & e^{i2\pi\frac{1}{|H|}} & \cdots & e^{i2\pi\frac{|H|-1}{|H|}} \\ & & \cdots & \\ 1 & e^{i2\pi\frac{(|H|-1)}{|H|}} & \cdots & e^{i2\pi\frac{(|H|-1)(|H|-1)}{|H|}} \end{bmatrix} \tag{86}$$

and left-multiply it by

$$B = \begin{bmatrix} 1 & 1 & \cdots & 1 \\ 1 & e^{-i2\pi\frac{1}{|H|}} & \cdots & e^{-i2\pi\frac{|H|-1}{|H|}} \\ & & \cdots & \\ 1 & e^{-i2\pi\frac{(|H|-1)}{|H|}} & \cdots & e^{-i2\pi\frac{(|H|-1)(|H|-1)}{|H|}} \end{bmatrix} \tag{87}$$

we obtain a matrix that is filled with $0$ except for each row $r$, one time the value $|H|$ at the column $c$ where $c$ such that $h_0^{-r}hh_0^c = e_H$, i.e. $hh_0^c = h_0^r$, this is exactly what $P_h$ represents, as we defined it such that if $hh_i = h_j$ then $P_h$ has a 1 at the column $j$ of its $i$-th row. Thus,

$$P_h = \frac{1}{|H|}B\psi_{h,j}C \tag{88}$$

Consider a block diagonal matrix $M$ where the first $|H|$ elements are all ones (on the diagonal), and then the matrix $B$ is repeated on the block diagonal $|A| - 1$ times.

$$M = \begin{bmatrix} 1 & & & & & & \\ & 1 & & & & & \\ & & \ddots & & & & \\ & & & 1 & & & \\ & & & & B & & \\ & & & & & B & \\ & & & & & & \ddots \\ & & & & & & & B \end{bmatrix} \overset{|H| \text{ times}}{\phantom{x}} \overset{|A|-1 \text{ times}}{\phantom{x}} \tag{89}$$

Consider $M'$ that is the repetition of $|A|$ times $\psi_{h,j}$

$$M' = \begin{bmatrix} 1 & & & & & & \\ & e^{i2\pi\frac{1}{|H|}j} & & & & & \\ & & \cdots & & & & \\ & & & e^{i2\pi\frac{|H|-1}{|H|}j} & & & \\ & & & & 1 & & \\ & & & & & e^{i2\pi\frac{1}{|H|}j} & \\ & & & & & & \cdots \\ & & & & & & & e^{i2\pi\frac{|H|-1}{|H|}j} \\ & & & & & & & & \cdots \end{bmatrix} \overbrace{\quad}^{\psi_{h,j}\ (\text{of order } |H|)} \quad (90)$$

$M'$ is of order $|A||H|$. Thus,

$$MM'C = \frac{1}{|H|}\begin{bmatrix} 1 & & & & & \\ & e^{i2\pi\frac{1}{|H|}j} & & & & \\ & & \cdots & & & \\ & & & e^{i2\pi\frac{|H|-1}{|H|}j} & & \\ & & & & P_h & \\ & & & & & P_h \\ & & & & & & \cdots \\ & & & & & & & P_h \end{bmatrix} \overbrace{\quad}^{|H|\ \text{elements}} \overbrace{\quad}^{(A-1)\ \text{times}} \quad (91)$$

which is of order $|A||H|$. Assuming the encoder takes care of the right-multiplication by $C$ and the scaling by $|H|$ (which it can learn to do), we do not write the scaling and right-multiplication by $C$.

The operator $\psi_{h,j,N}$ is $\frac{N}{|H|}$ repetitions of $\psi_{h,j}$, hence it can be seen as $\frac{N}{|A||H|}$ repetitions of $M'$. Thus, let us denote $Q$ a $N$ by $N$ block diagonal matrix form of $\frac{N}{|A||H|}$ repetitions of $M$. When multiplying $Q$ with $\psi_{h,j,N}$ we can get $MM'$ repeated $|A||H|$ times:

$$Q\psi_{h,j,N} = \begin{bmatrix} 1 & & & & & & & \\ & e^{i2\pi\frac{1}{|H|}j} & & & & & & \\ & & \cdots & & & & & \\ & & & P_h & & & & \\ & & & & \cdots & & & \\ & & & & & P_h & & \\ & & & & & & 1 & \\ & & & & & & & e^{i2\pi\frac{1}{|H|}j} \\ & & & & & & & & \cdots \\ & & & & & & & & & P_h \\ & & & & & & & & & & \cdots \end{bmatrix} \overbrace{\quad}^{MM'} \overbrace{\quad}^{MM'} \quad (92)$$

This is the matrix that is multiplied with $\psi_{y,k',N}L_2\psi_{x,k,N}$. However, the order of the rows in the resulting does not correspond to the ordering of the characters in $\psi_{y,k',N}L_2\psi_{x,k,N}$. If we want to match the operator in Equation 66 such that

$$\psi_{a,h,N}z = (\psi_{y,k',N}L_2\psi_{x,k,N}L_1\psi_{h,j,N})z \quad (93)$$

We need $L_1$ be the product of two matrices $P$ and $Q$, such that $P$ reorders the rows of the vector $Q\psi_{h,j,N}z$ to match the ordering of the characters in $\psi_{y,k',N}L_2\psi_{x,k,N}$. The result of

$$(\psi_{y,k',N}L_2\psi_{x,k,N})PQ\psi_{h,j,N}z \quad (94)$$

will be a vector that is a permuted version of the vector we would get with $\psi_{a,h,N}z$, and a linear decoder can learn to reorder it if we want to exactly match $\psi_{a,h,N}z$.

## D.6 DETAILS OF SECTION D.4

### D.6.1 STABILIZERS OF ORBITS CASE 2

We show here that for $\chi_r = \chi_{(x_1,y_1)}(x,y)$ with $x_1 \neq 0$ or $y_1 \neq 0$ the stabilizer group $H_i$ is only $\{e_H\}$.

*Proof.* Let us consider the stabilizer group of $\chi_{(x_1,y_1)}$. It is composed of the elements $h \in H$ such that for $a \in A$, $h_{\chi_{(x_1,y_1)}}(a) = \chi_{h((x_1,y_1))}(a) = \chi_{(x_1,y_1)}(a)$. In other words, we must have that the character of the rotated vector $h((x_1,y_1))$ is the same as the character corresponding to $(x_1,y_1)$, for any $a$. Seeing characters as 2D complex numbers, we must have $(x_1,y_1) = (\cos\theta x_1 - \sin\theta y_1, \sin\theta x_1 + \cos\theta y_1)$, i.e. the rotated vector corresponding to the character is equal to itself. As we employ boundary conditions, this is for example the case for $\theta = \pi$ or $\theta = -\pi$ for $x_1 = \frac{K}{2}, y_1 = \frac{K'}{2}$, since the inverse of $(\frac{K}{2}, \frac{K'}{2})$ is itself. But we restrict ourselves and we consider only odd $K$ and $K'$, such that there is no angle $\pi$ such that the result of the rotation of $(x_1,y_1)$ gives the same vector in complex space. $\qquad\square$

### D.6.2 DEGREES OF THE REPRESENTATIONS

**Case 1** In Case 1, we have $|H|$ representations of degree 1.

**Case 2** Recall that in Case 2, $H_i = \{e_H\}$ and we use $\rho = 1$, thus the character of the induced representation is

$$\chi_{Ind_{G_i}^G(\chi_r \otimes 1)}(a,h) = \sum_{h' \in H\,s.t.\,h'^{-1}hh'=e_H} \chi_r(h(a))\chi_1(e_H) \tag{95}$$

$h'^{-1}hh' = e_H$ means $h'^{-1}h = h'^{-1}$. If $h = e_H$, $h'$ will span the entire $H$ and we have

$$\chi_{Ind_{G_i}^G(\chi_r \otimes 1)}(a,e_H) = \sum_{h' \in H} \chi_r(a)\chi_1(e_H) = |H|\chi_r(a) \tag{96}$$

If $h \neq e_H$ then there is no element $h'$ for which this is true. Indeed, otherwise it means $h = h'h'^{-1} = e_H$ leading to a contradiction. Thus:

$$\chi_{Ind_{G_i}^G(\chi_r \otimes 1)}(a,h) = \begin{cases} |H|\chi_r(a) \text{ if } h = e_H \\ 0 \text{ if } h \neq e_H \end{cases}$$

To obtain the degree of the representation, we calculate the character of the identity element.

$$\chi_{Ind_{G_i}^G(\chi_r \otimes 1)}(e_A,e_H) = \sum_{h' \in H\,s.t.\,h'^{-1}e_H h'=e_H} \chi_r(e_A)\chi_1(e_H) \tag{97}$$

$$= \sum_{h' \in H} \chi_r(e_A)\chi_1(e_H) = |H|\chi_r(e_A)\chi_1(e_H) = |H|\chi_1(e_H) = |H| \tag{98}$$

where we use the fact $\chi_r(e_A) = 1$ ($A$ is abelian hence its irreducible characters are of degree 1) and that $h'^{-1}e_H h' = e_H$ is true for all $h' \in H$. Hence, the degree of the induced representation is $|H|\chi_1(e_H) = |H|$ since $\chi_1(e_H) = |\{e_H\}| = 1$.

Thus in case 2 each orbit induces a unique induced representation of degree $|H|$, and we have $\frac{|A|-1}{|H|}$ orbits in the second case. This shows we derived all the irreducible representations of $G$, as if we do the sum of the degree squared of irreducible representations we have

$$|H| + \frac{|A|-1}{|H|}|H|^2 = |H| + |A||H| - |H| = |A||H| \tag{99}$$

which correctly equals the order of $G$, see Scott & Serre (1996, Corollary 2).

# E    ADDITIONAL RESULTS

## E.1    QUANTIFICATIONS

**Test Mean Squared Error**    Table 2 reports test MSE for disentangled, supervised shift and weakly supervised shift operators.

Table 2: Test mean square error (MSE) $\pm$ standard deviation of the mean over random seeds. Numbers in () refer to the number of rotations, the number of translations on the $x$-axis, and the number of translations on the $y$-axis in this order. The case of multiple transformation does not apply to the weakly supervised shift operator, and we did not experiment on translated MNIST with the weakly shift operator as its performance on translated simple shapes and rotated MNIST were showing its relevance already.

| Dataset | Model | | |
|---|---|---|---|
| | Disentangled | Shift | Weak. sup. shift ($K_L = 10$) |
| Shapes (10,0,0) | $0.0208 \pm 5.2\mathrm{e}{-6}$ | $\mathbf{0.0002 \pm 1.8e-6}$ | $0.001 \pm 1.3\mathrm{e}{-3}$ |
| Shapes (0,10,0) | $0.0352 \pm 9.0\mathrm{e}{-6}$ | $\mathbf{0.0052 \pm 9.6e-6}$ | $0.0097 \pm 5.2\mathrm{e}{-3}$ |
| Shapes (0,0,10) | $0.0353 \pm 7.6\mathrm{e}{-6}$ | $\mathbf{0.0052 \pm 1.5e-5}$ | $0.0115 \pm 4.9\mathrm{e}{-3}$ |
| Shapes (0,5,5) | n/a | $0.0047 \pm 1.9\mathrm{e}{-5}$ | n/a |
| Shapes (4,5,5) | n/a | $0.0049 \pm 7.7\mathrm{e}{-6}$ | n/a |
| Shapes (5,5,5) | n/a | $0.0021 \pm 5.9\mathrm{e}{-6}$ | n/a |
| MNIST (10,0,0) | $0.0660 \pm 8.2\mathrm{e}{-5}$ | $\mathbf{0.0004 \pm 5.6e-6}$ | $0.0035 \pm 3.9\mathrm{e}{-3}$ |
| MNIST (0,10,0) | $0.0838 \pm 3.2\mathrm{e}{-5}$ | $\mathbf{0.0079 \pm 5.1e-5}$ | n/a |
| MNIST (0,0,10) | $0.0857 \pm 5.1\mathrm{e}{-5}$ | $\mathbf{0.0062 \pm 3.2e-5}$ | n/a |
| MNIST (4,5,5) | n/a | $0.004 \pm 4.0\mathrm{e}{-5}$ | n/a |

**LSBD Disentanglement Measure**    We compute the LSBD disentanglement metric to quantify how well the shift operator captures the factors of variation compared to the disentangled operator (see Anonymous (2021)). Note traditional disentanglement metrics are not appropriate as they describe how well factors of variation are restricted to subspaces in contrast to our proposed framework using distributed latent operators. LSBD, on the other hand, measures how well latent operators capture each factor variation to quantify disentanglement even for distributed operators. Using LSBD, we quantify the advantage of the shift operator with LSBD of 0.0020 versus the disentangled operator with LSBD of 0.0106 for the models in Fig. 3.A and 3.B.

## E.2    ADDITIONAL ANALYSES

**Latent Variance and PCA Analysis**    The analysis in Fig. 1.C and 1.D quantitatively measures disentanglement in the latent representation for VAE and variants on rotated MNIST. We first apply every transformation to a given shape and compute the variance of the latent representation as we vary the transformation. The final figure shows the average variance across all test samples. For the PCA analysis, we seek to determine whether the transformation acts on a subspace of the latent representations. We first compute the ranked eigenvalues of the latent representations of each shape with all transformations applied to the input. We then normalize the ranked eigenvalues by the sum of all eigenvalues to obtain the proportion of variance explained each latent dimension. Finally, we plot the average of the normalized ranked eigenvalues across all test samples.

**Additional latent traversals**    Figures 5 and 6 show all latent traversals for the model with the best validation loss. We see the success of disentanglement in the case of a single digit and the failure of a latent to capture rotation in the case of multiple digits. We show a more granular traversal with 50 plots per latent for each baseline in 7.In Figures 9, 10 we show comparable results for the case of translations along the x-axis and y-axis.

**Additional results for the distributed operator models**    Fig. 14 shows additional results on MNIST. Fig. 14a shows that the weakly supervised shift operator performs well on Rotated MNIST, and in Fig. 14b we see the stacked shift operator model is able to correctly encode the multiple

transformations case on MNIST. Fig. 15 shows performance of the weakly supervised shift operator on translated simple shapes (either $x$ or $y$ translations). The effect of the number of latent transformations is explored in App. B.3. It shows the best model is in the case of translation obtained with 21 transformations. Nonetheless, to be able to feed to the model the correct 10 transformations needed in these plots, the reported plots are for the model with 10 latent transformations. Fig. 16 shows example results for the case of the Special Euclidean group (i.e. rotations in conjunction with translations) in a setting where the semi-direct product structure is broken (see Appendix D.4.1). Here, we used a rotation group of order 5. We see the stacked shift operator model performs well nonetheless.

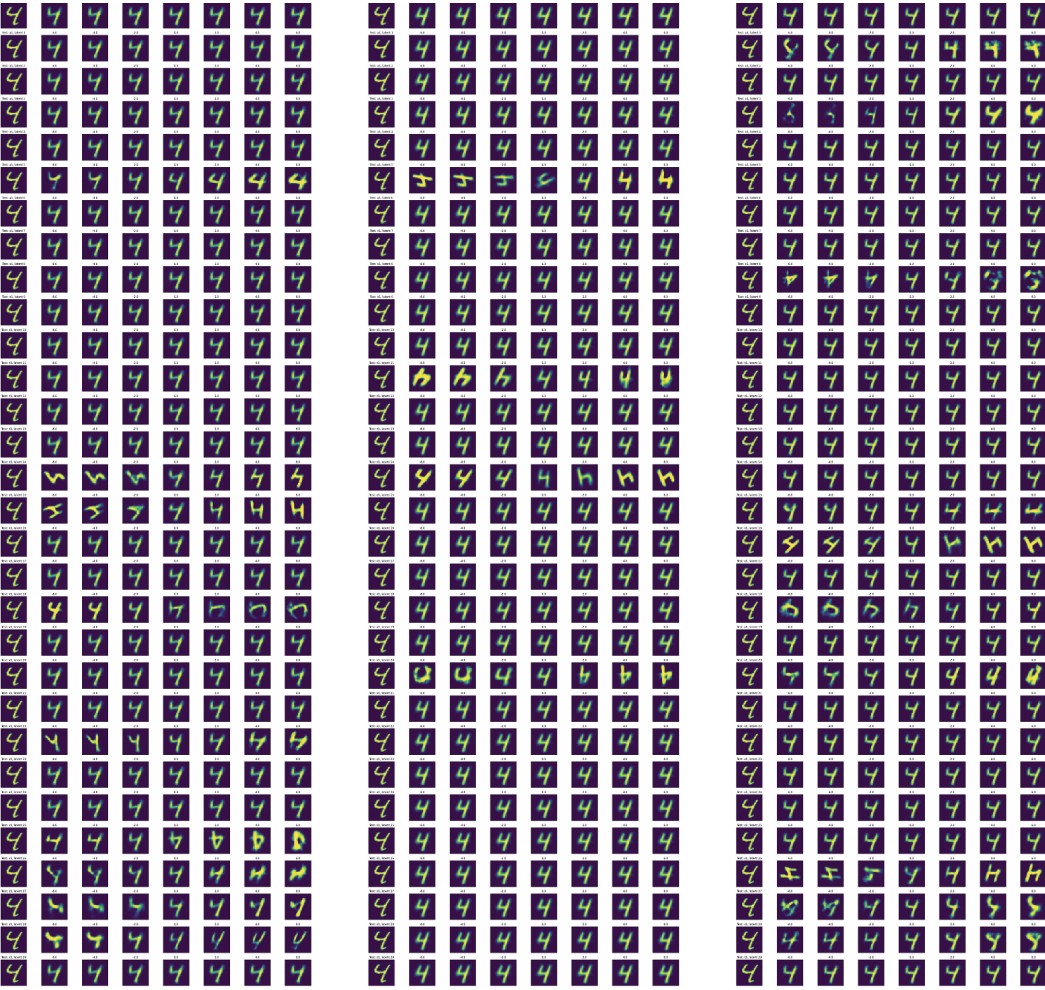

Figure 5: **Single Rotated MNIST Digit Label**: Latent traversals for VAE (left), $\beta$-VAE (middle), CCI-VAE (right) trained on a single rotated MNIST digit (10 rotations). Latent traversal spans the range $[-6, 6]$ for each latent dimension.

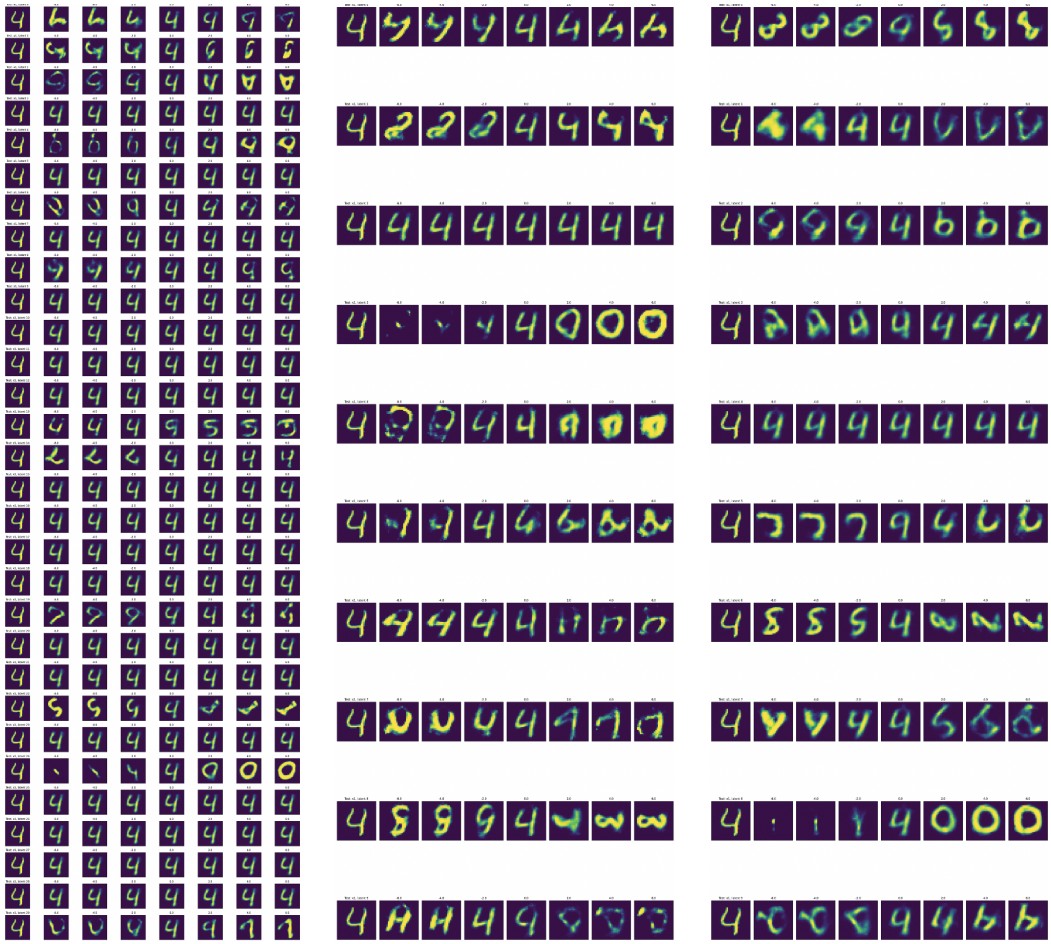

Figure 6: **Rotated MNIST**: Latent traversals for VAE (left), $\beta$-VAE (middle), CCI-VAE (right) trained on all rotated MNIST digits (10 rotations). Latent traversal spans the range $[-6, 6]$ for each latent dimension. Note in this case, the best validation model for VAE contained 30 latent dimensions whereas $\beta$-VAE and CCI-VAE contain 10.

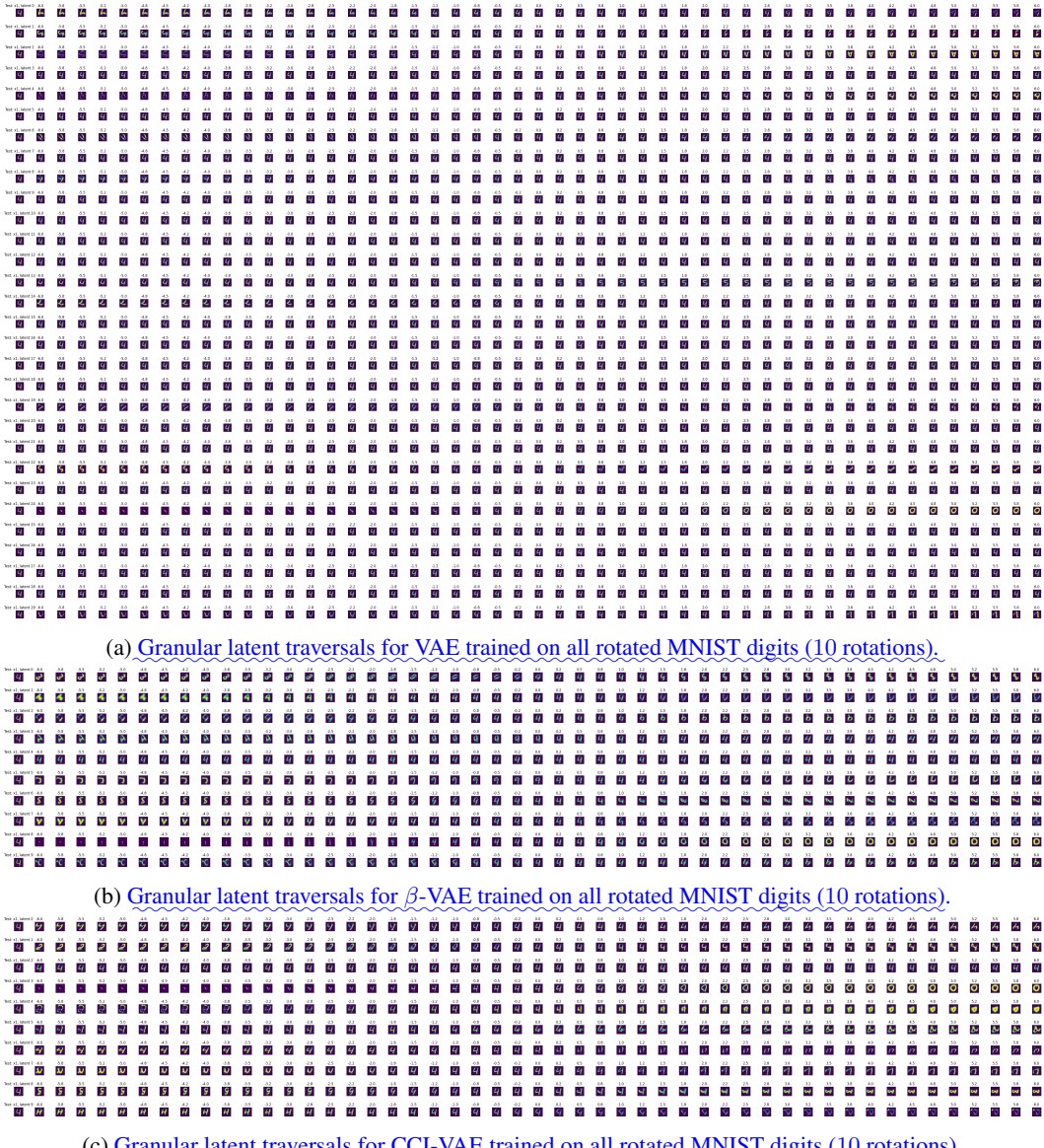

(a) Granular latent traversals for VAE trained on all rotated MNIST digits (10 rotations).

(b) Granular latent traversals for $\beta$-VAE trained on all rotated MNIST digits (10 rotations).

(c) Granular latent traversals for CCI-VAE trained on all rotated MNIST digits (10 rotations).

Figure 7: Granular latent traversals for VAE $\beta$-VAE, CCI-VAE trained on all rotated MNIST digits (10 rotations). Latent traversal spans the range $[-6, 6]$ with step size $0.25$ yielding 50 plots per latent dimension. Note in this case, the best validation model for VAE contained 30 latent dimensions whereas $\beta$-VAE and CCI-VAE contain 10

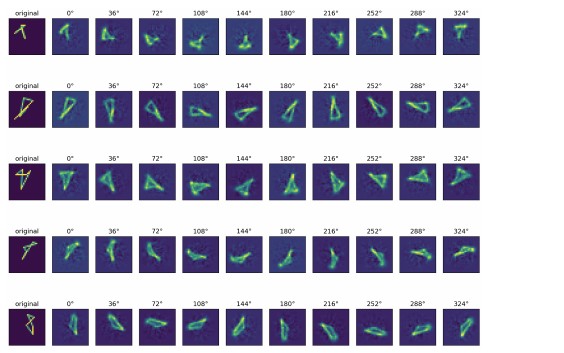

(a) Supervised disentangled operator on Rotated Shapes (10 rotations)

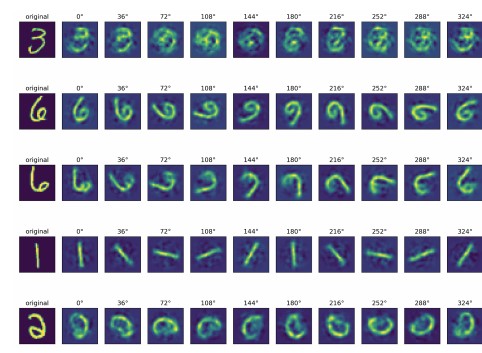

(b) Supervised disentangled operator on Rotated MNIST (10 rotations).

Figure 8: Non-linear disentangled operator with latent rotations

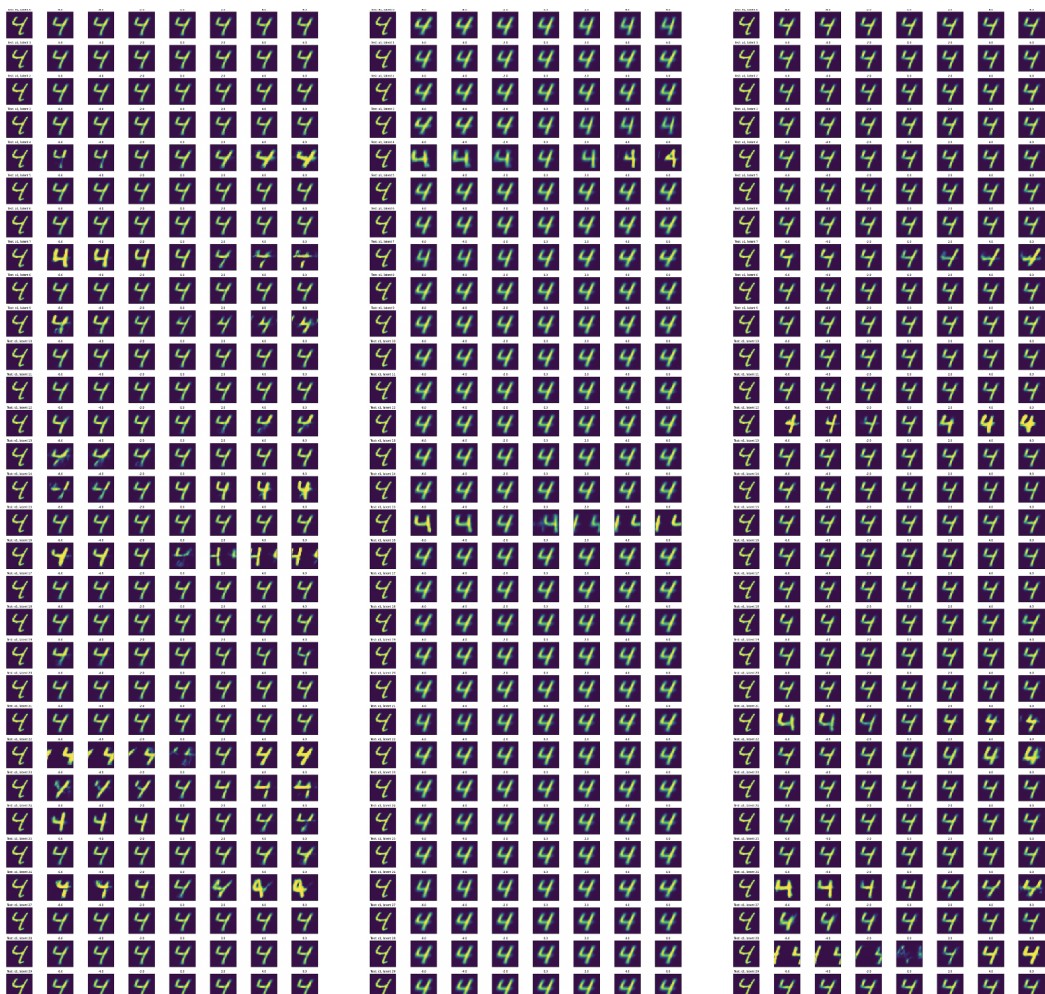

Figure 9: **Single MNIST Digit Label Translated along X-Axis**: Latent traversals for VAE (left), $\beta$-VAE (middle), CCI-VAE (right) trained on a single MNIST digit translated along the $x-$axis (10 translations). Latent traversal spans the range $[-6, 6]$ for each latent dimension.

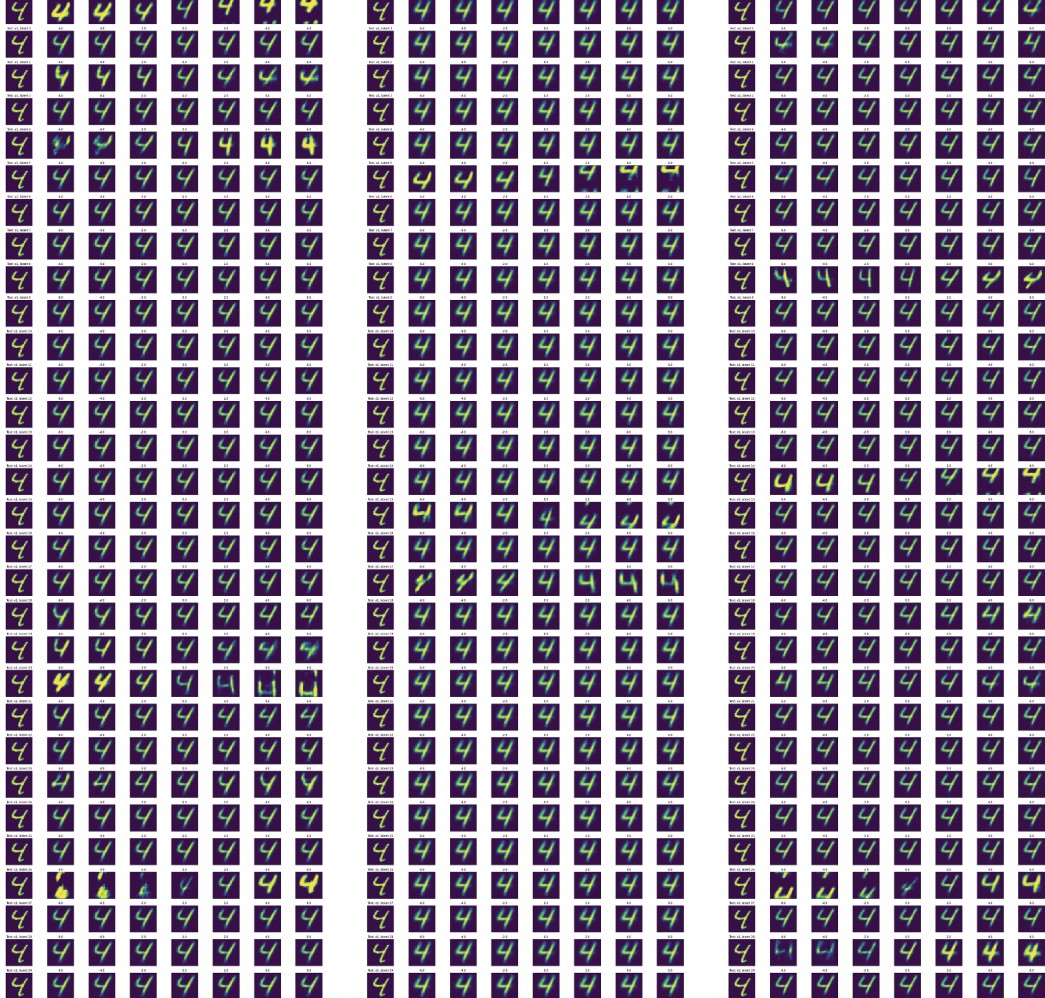

Figure 10: **Single MNIST Digit Label Translated along Y-Axis**: Latent traversals for VAE (left), $\beta$-VAE (middle), CCI-VAE (right) trained on a single MNIST digit translated along the $y-$axis (10 translations). Latent traversal spans the range $[-6, 6]$ for each latent dimension.

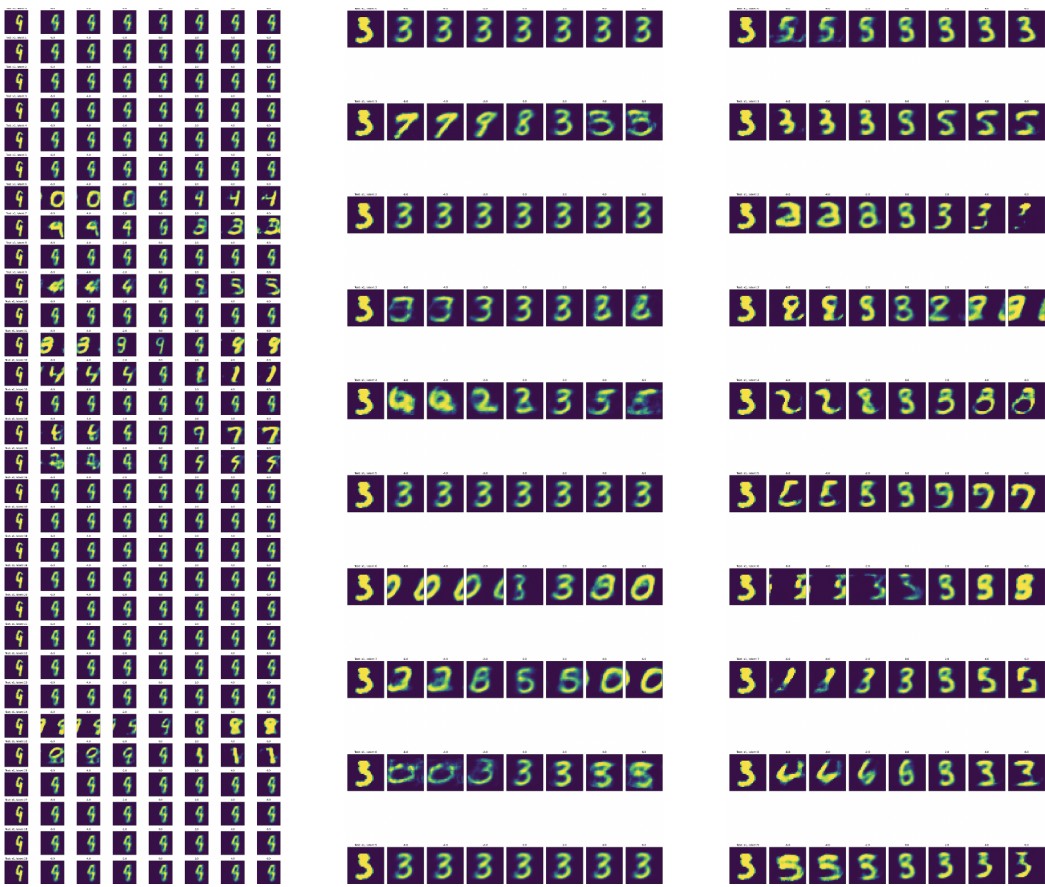

Figure 11: **MNIST Translated along X-Axis**: Latent traversals for VAE (left), $\beta$-VAE (middle), CCI-VAE (right) trained on all MNIST digits translated along the x-axis (10 translations). Latent traversal spans the range $[-6, 6]$ for each latent dimension.

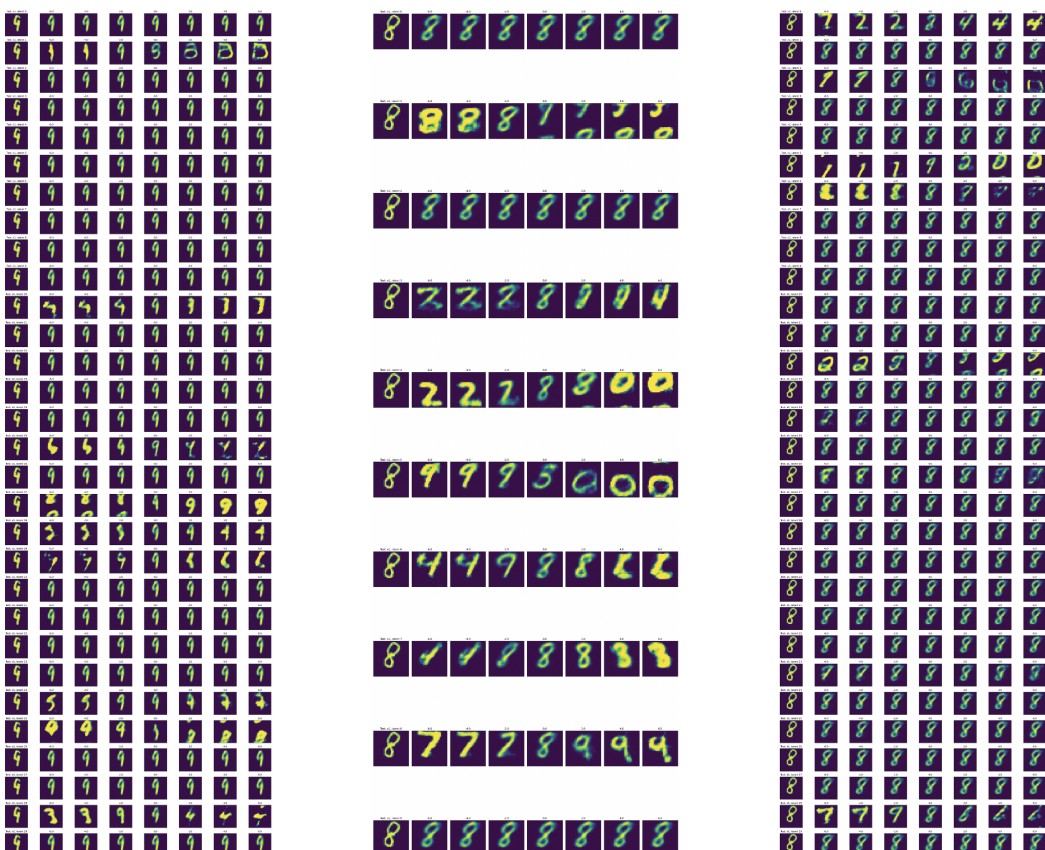

Figure 12: **MNIST Translated along Y-Axis**: Latent traversals for VAE (left), $\beta$-VAE (middle), CCI-VAE (right) trained on all MNIST digits translated along the y-axis (10 translations). Latent traversal spans the range $[-6, 6]$ for each latent dimension.

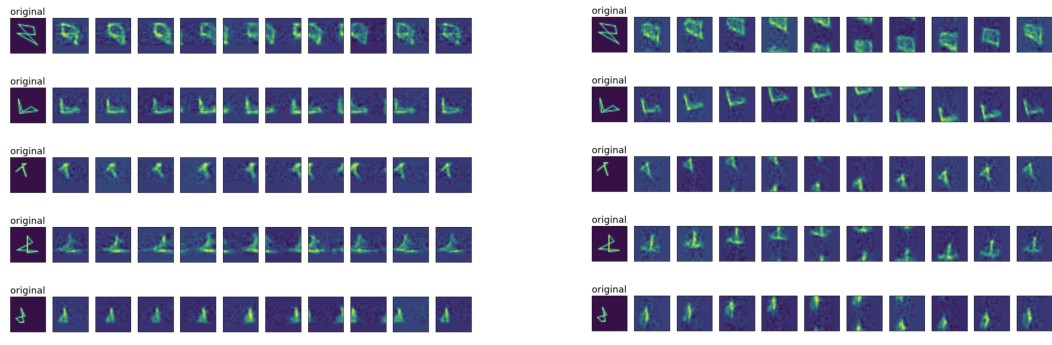

(a) Supervised disentangled operator on translated shapes along the x-axis (10 translations)

(b) Supervised disentangled operator on translated shapes along y-axis (10 translations).

Figure 13: Non-linear disentangled operator with latent translations

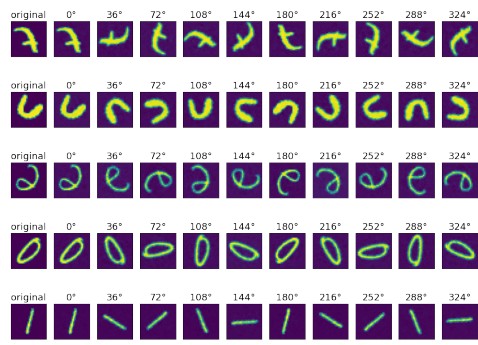

(a) Weakly supervised shift operator on Rotated MNIST (10 rotations).

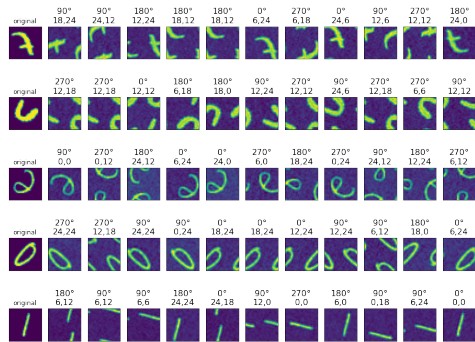

(b) Supervised shift operator on Rotated-Translated MNIST (4 rotations, 5 $x$-translations and 5 $y$-translations).

Figure 14: MNIST additional experiments.

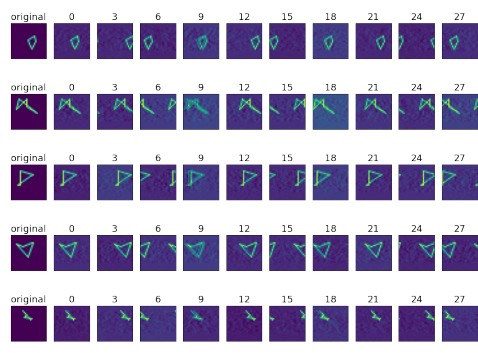

(a) Weakly supervised shift operator on $x$-translations (10 translations).

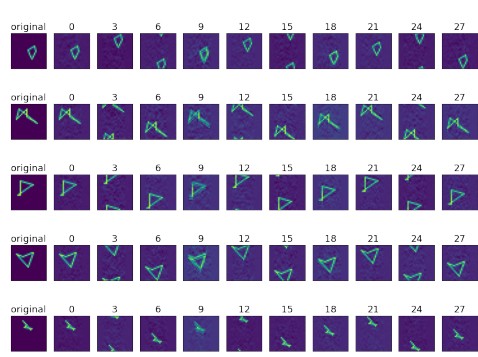

(b) Weakly supervised shift operator on $y$-translations (10 translations).

Figure 15: Simple shapes additional experiments.

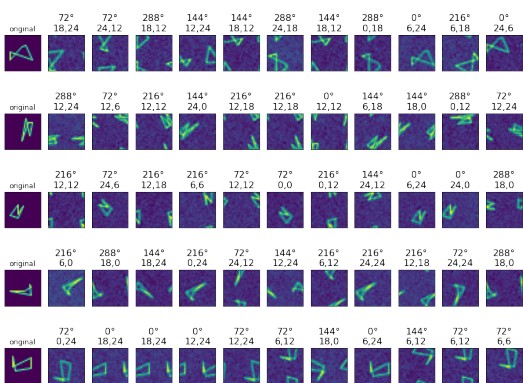

Figure 16: Supervised shift operator on Rotated-Translated simple shapes when the semi-direct product structure is not respected as rotations angles are $j\frac{\pi}{5}$, $j = 1, \ldots 5$ (5 rotations, 5 $x$-translations and 5 $y$-translations).

Test Reconstruction

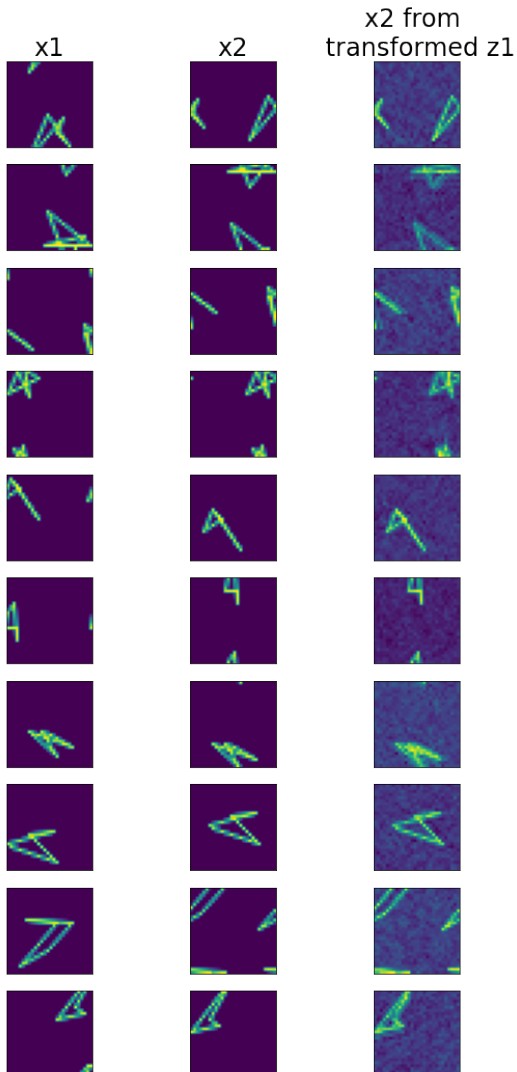

Figure 17: Pairs of test samples and their reconstructions for the stacked shift model with 5 translations in both $x$ and $y$.

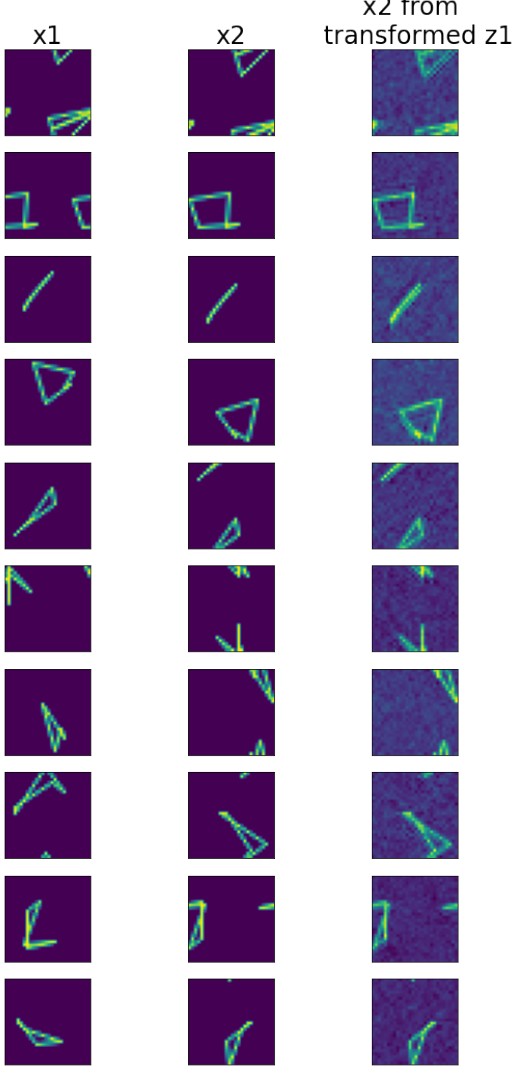

Figure 18: Pairs of test samples and their reconstructions for the stacked shift model with $4$ rotations and $5$ translations in both $x$ and $y$.

