# OpenReview forum: "Addressing the Topological Defects of Disentanglement"
_ICLR.cc/2021/Conference — Reject_

### Official Review · AnonReviewer2 · 2020-10-21

**Rating:** 5
**Confidence:** 4

**Review:**

This paper studies the notion of disentanglement in a group representation theoretic setting. Disentangling is sometimes conceptualized as mapping distinct factors (e.g. position / orientation) to distinct subspaces. It is shown theoretically that such a naive notion of disentangling is impossible for topological reasons, and this is confirmed empirically. An alternative definition of disentanglement is given, where instead of confining the effect of each transformation to a subspace, an operator is used that acts on the whole latent space (this operator is chosen as a shift operator, which works for cyclic groups). It is shown empirically that an autoencoder with a shift operator in latent space is better able to learn rotations and translations.

The paper does a good job explaining why the naive notion of disentangling leads to topological problems, and convincingly backs this up with experiments as well. The insight is not new to me personally, but I can't find a reference that explains it and I think it is not widely understood, so I consider this an important contribution to the (very muddled) discourse on disentangling.

Definition 1 provides a new definition of disentangling. However, the statement is not very precise, and I am not convinced that it can reasonably be considered as a definition of disentanglement. The definition is:

"A representation is said to be disentangled with respect to a particular decomposition of a symmetry group into subgroups, if there is a family of known operators acting on this representation, potentially distributed across the full latent, where each operator is equivariant to the action of a single subgroup."

Based on the rest of the paper, I think this means that we have for each subgroup G_i an operator phi_i(g) acting on the latent space. The definition does not make it clear that we wish the encoder to be equivariant wrt this operator and some operator acting on the input space, but I will assume that is what is meant (otherwise, having an operator acting on the latent space is a rather vacuous requirement on the encoder/representation). The definition does speak of the operator being equivariant, which I will take to mean that it is a group representation, i.e. phi(gg') = phi(g)phi(g'). The operator being distributed I will take to mean that phi(g) can be any linear map, not necessarily acting trivially on a subspace or being (block-) diagonal / reduced.

The definition mentions that each subgroup should have its own operator, but since all of them act on the whole subspace this seems to a trivial constraint. Indeed if we have a representation of the whole group acting on the latent space, simply restricting it to each subgroup gives us a representation of the subgroups. I would further note that what is done in practice in the paper is different from this definition, because we have one latent space per operator, not multiple operators acting on the same space.

Under this interpretation, I don't see how the definition is saying anything else than that the network should be equivariant wrt some representation of the group acting on the input and output space. Although equivariance is a good property for various reasons, it does not seem to me to be reasonable definition of disentangling by itself. Indeed, the identity map satisfies this constraint trivially.

It may be that I have misunderstood definition 1, but this strengthens the case for making it mathematically precise.

Even if one can question whether Def 1 is a good formalization of disentangling, the paper does show empirically that it is easier to learn an equivariant encoder/decoder when the latent operator is a shift operator or a diagonalized complex version of it, rather than a disentangled operator (with one 2x2 rotation matrix block and an identity block; fig 3b). Although I don't know if these two approaches have been compared before, several older papers consider similar models to the shift operator model.

For instance, in a sequence of papers Memisevic & Hinton considered factorized RBMs that do something similar. Cohen & Welling described a representation-theoretic version of this model which is very similar what is presented in this paper (at least the linear AE), and also gave a definition of disentangling (under this definition, the complex diagonal shift operator is disentangled while the original shift operator is not). Models with a stack of multiple operators were considered by Sohl-Dickstein et al.

If one wishes to define a notion of disentangling based on subgroups and representations, it may be worth investigating subgroup adapted / Gelfand-Tsetlin bases.

In summary, I think this paper contains several interesting observations and results, and I think the general direction is very interesting and deserves further study. However, I'm not convinced that this paper provides a good definition of disentangling, the experiments although convincing and well executed are restricted to simplified domains, and some of the insights / methods presented in the paper are already present in earlier work. Nevertheless I hope the authors will not be discouraged, and continue to work on this important and fundamental problem using the tools of representation theory.

References
Memisevic & Hinton, Learning to Represent Spatial Transformations
with Factored Higher-Order Boltzmann Machines, 2010
Sohl-Dickstein, Wang, Olshausen, An unsupervised algorithm for learning Lie group
transformations, 2010
Cohen & Welling, Learning the Irreducible Representations of Commutative Lie Groups, 2014
Wakin, Donoho, Choi, Baraniuk, The multiscale structure of non-differentiable image manifolds, 2005

----
Post-discussion update:
Having read the other reviews, author response and updated paper, I still think this paper is borderline. The insight that disentangling transformations as naively defined is impossible for topological reasons is valid and interesting, but seems to have been already observed by others, e.g. Falorsi et al. Nevertheless the paper does a good job explaining this so it could be useful, as some authors seem to not know about this issue. The definition of disentangling still seems a bit vague to me, and I'm not convinced of practical applicability of the proposed method.

---

> ### Author Response · Authors · 2020-11-18
> **Response to Reviewer 2**
>
> _1) Unclear definition of disentanglement + identity map would trivially satisfy equivariance_
>
> We thank the reviewer for his insightful remarks which allowed us to conceptually clarify our proposed definition of disentanglement. The reviewer is correct to point out that a consequence of our definition of disentanglement is that the network should be equivariant wrt to some representations of the group acting respectively on the input and latent space. However, our proposed definition goes beyond the simple requirement of equivariance (and we now clarify this definition further in the main text). Indeed, an important additional requirement of our definition is that the operators acting on the latent space should be controllable, in the sense that representations should be manipulable at test time to emulate the transformations learned. This desiderata of controllability can be achieved either by choosing these operators in advance (i.e. hard-coded operators), or by learning their explicit form (see Connor and Rozell for an example of learned operators). In section 4, we show how the operators can be chosen in advance in the case of simple affine transformations acting on the input space.
>
> => We reformulated our definition of disentanglement (Def 1) for conceptual clarity. The definition more clearly connects the notion of an equivariant model with controllable distributed operators with the notion of disentanglement. We also included a more precise definition of equivariance in the main text (Section 3.3). We also included additional background on the use of group theory for describing transformations in Section 3.4.
>
>
> _2) The definition mentions that each subgroup should have its own operator, but since all of them act on the whole subspace this seems to be a trivial constraint. How is it different than having a representation for the entire group and restrict it to each subgroup?_
>
> Indeed, if a representation for the entire group is known in advance, it can very well be restricted to subgroups. However, choosing an operator in latent space capturing the entire group structure requires knowledge of the group structure a priori such that we can correctly identify the representation operator for this group (e.g. we derive the representation operator in Appendix D.4 for the discrete finite Special Euclidean Group). In this case, one could then obtain the representation for each element of each subgroup by plugging in adequate values in the variables of the operator. However, this option requires a priori knowledge of the group and its representation, and thus lacks flexibility. Using different operators for each subgroup, stacked together with intermediate layers, as in our stacked shift operator model, one can at the same time (i) control the representation learned for each subgroup after training and (ii) flexibly learn such operators without deriving the form of the operator for the entire group a priori. Note in this case that it is necessary to include intermediate linear layers, as the different operators should not commute in the case of non-commutative groups.
>
>
> _3) I would further note that what is done in practice in the paper is different from this definition, because we have one latent space per operator, not multiple operators acting on the same space._
>
> In the specific scenario we consider, where we choose the operator in advance to be the shift operator, we cannot have all operators corresponding to all affine transformation simultaneously act on the same latent space (otherwise the operators would all map to the same transformation since they all have identical form). This is why we propose an alternative solution consisting in stacking layers, so that each shift operator acts separately on its own latent space, which is learned to map the operator to one of the transformations present in the data. We however want to emphasize that stacking is only one possible implementation of disentanglement consistent with our definition, but in other cases, for example the case where these operators are learned, the operators could all be acting in a common latent space, as in Connor and Rozell for example.
>
> => We added a discussion point about how the operators could be learned in a common latent space, as opposed to be stacked in different layers.

---

> > ### Author Response · Authors · 2020-11-18
> > **Response to Reviewer 2 (continued)**
> >
> > _4)  Additional references + Novelty concerns_
> >
> > We thank the reviewer for these interesting references that we were not familiar with. We agree that we are not the first to propose disentanglement via distributed operators, but we believe that we are the first to define distributed disentanglement formally and clearly oppose it to traditional disentanglement via subspaces. We also believe that we are the first to justify theoretically the use of distributed disentanglement by showing the failure mode of traditional disentanglement using arguments from topology, and that we are the first to motivate the use of the shift operator to deal with affine transformations using arguments from character theory.
> >
> > =>We now cite Memisevic & Hinton, Cohen & Welling, Sohl-Dickstein et al in our paper.

---

> > > ### Comment · AnonReviewer2 · 2020-11-23
> > > **Not yet convinced**
> > >
> > > It's nice to see the authors more fully embracing a representation theoretic perspective in the new version of the paper, but I am still not convinced that the definition of disentangling given is novel, meaningful and precise. Firstly, the notion of "controllable operators" is still quite vague. I don't think there is a mathematical criterion for when an operator is "controllable". Rather it seems to have something to do with the operator being known or computable by the researcher, but it would be strange to say the disentangledness of the representation depends on whether we know/can compute the operators acting on it. A representation could be disentangled without us knowing in what way it is / what the operators are.
> > >
> > > The claim that the proposed approach/definition enables learning without knowing the group seems to be based on a confusion between the concepts of (abstract) group and group action. It is stated that "Importantly, the shift operator does not require knowledge of the transformation in advance, only the cycle order of each group". But since the method only works for cyclic groups and we have to know the cycle order, we essentially have to know the group. There is only one cyclic group of order k, up to isomorphism. This same group may act in very different ways on the input space, e.g. by rotations or by cyclic translations. So one could say that the method does not require computing explicitly the representation matrices of the group in the input space, although it does require pairs of inputs related by transformations g acting via that representation. But this isn't really new, and it's unclear to me how this is related to the definition of disentangling.

---

> > > > ### Comment · AnonReviewer4 · 2020-11-23
> > > > **I agree**
> > > >
> > > > Since I asked very similar questions in my review (#4), I agree with reviewer 2 that your response (see https://openreview.net/forum?id=cbdp6RLk2r7&noteId=JFX45HKEhFK above) is not convincing. To know that the group is cyclic with a particular cycle order, one needs a lot of prior information about the task. Moreover, what happens if the group of interest is non-cyclic?
> > > >
> > > > I don't necessarily consider this problem as a showstopper for the paper (as reviewer 2 might), but a better discussion of its implications would definitely be a plus.

---

> > > > > ### Author Response · Authors · 2020-11-24
> > > > > **Response to Rev 2 and 4**
> > > > >
> > > > > We thank Rev 2 and 4 for taking the time to thoroughly engage with our work, which helps us clarify our contributions further here and in the paper.
> > > > >
> > > > > _1) The proposed alternative definition of disentanglement is not novel or meaningful_
> > > > >
> > > > > We understand the concerns of Rev 2 about the novelty and meaningfulness of our definition of disentanglement. We would first like to take a step back and emphasize that the main contribution of our paper is not to propose a new definition of disentanglement. Instead, it is to show that disentanglement introduces topological defects in the encoder for a large family of transformations. We then argue that one can achieve the same *desiderata* as disentanglement ---namely to identify and isolate the transformations present in the data--- by learning to map each of these transformations to a different operator in latent space. As pointed out by the reviewer, this strategy is well known and already used in the literature, and is usually referred to as "learning transformations" (e.g. Connor and Rozell, Dupont et al).  However this prior work does not justify theoretically the choice of distributed operators in latent space. The main contribution of our work is to justify *why* distributed operators should be used instead of disentangling operators (=> to avoid topological defects). We believe---as most of the reviewers---that we are the first to provide a theoretical motivation for this choice and that this insight is a valuable contribution on its own. In particular, we note that Reviewer 2 acknowledges the importance of this finding despite the fact that he already knew it himself.
> > > > >
> > > > > => In order to emphasize our main contribution more clearly, we propose to reorganize the paper as follows:
> > > > > 1. Empirical Limitations of Disentanglement (unchanged)
> > > > > 2. Topological Defects of Disentanglement (unchanged, current section 3.1 and 3.2)
> > > > > 3. Learning Transformations with Distributed Operators as an alternative to Disentanglement (current sections 3.3, 3.4 and 4)
> > > > > - 3.1 Definition of ‘Learning Transformations’  as an alternative to Disentanglement (see definition below)
> > > > > - 3.2 Illustration of this strategy in the simple case of affine transformations: representation theory guarantees the success of the shift operator to learn any affine transformation (unchanged)
> > > > > - 3.3 Illustration of this strategy in the simple case of affine transformations: we verify empirically that we can learn rotations, translations and combinations thereof using the shift operator in latent space, but not the disentangled operator (results unchanged, text reformulated)
> > > > >
> > > > > Instead of redefining disentanglement, we will simply propose in the new section 3 to learn transformations with distributed operators as an alternative to disentanglement (with the same desiderata in mind of isolating the factors of variation in data). We define below what is meant by learning transformations:
> > > > >
> > > > > "Learning transformations consists in finding an invertible encoder $f$ and a family of operators $\phi_k$ in latent space, such that each operator corresponds to a subgroup acting on image space and the resulting model is equivariant to the group of transformations.  Learning transformations can either be achieved by (1) hard-coding the operators and learning the encoder/decoder parameters from examples of transformed images (but necessitates a priori knowledge of the transformation) (2) learning jointly the operators and the encoder/decoder parameters (e.g. Connor and Rozell).
> > > > > Learning transformations achieves the same desiderata as disentanglement, namely to isolate factors of variation acting on the data, only in separate operators rather than separated latent dimensions. "
> > > > >
> > > > > We will explicitly say in the paper that this definition is not new and cite the relevant literature.
> > > > >
> > > > > => We thank again the reviewer for his relevant comments, his involvement in his review and the discussion, and we hope that the proposed reorganization of the paper clarifies our main contribution and addresses the novelty concerns of Rev 2. We would be happy to use this reformulation of our claims upon acceptance, and we are open to additional suggestions of the reviewers to improve the clarity of our contributions further.

---

> > > > > > ### Author Response · Authors · 2020-11-24
> > > > > > **Response to Rev 2 and 4 (continued)**
> > > > > >
> > > > > > _2) The hard-coded shift operator proposed only works for cyclic groups_
> > > > > >
> > > > > > We agree that the setup presented in Section 4 (Distributed Disentanglement in Practice) only allows to learn (1) cyclic groups and (2)  combinations of cyclic groups (e.g. direct product of cyclic groups like for example translations in x and y, or semi-direct product of cyclic groups like for example rotations + translations).
> > > > > >
> > > > > > However, we would like to put this limitation of our work in perspective by re-emphasizing here the main contribution of the paper and the goal of Section 4. The experiments in Section 4 are not the main contribution of the paper. Our main contribution is to prove that traditional disentanglement via subspaces introduces topological defects in the encoder for a large family of transformations---a contribution the reviewers think is valid and important. We then propose to learn transformations with distributed operators in latent space, because it is an alternative to disentanglement which resolves these topological defects while preserving the desiderata of isolating the factors of variation in data.  We prove using representation theory that, at least in the case of affine transformations, we can find distributed operators which satisfy the topology of the problem (the shift operator). The only goal of Section 4 is to provide empirical evidence for the theoretical finding that an affine transformation in image space can be mapped to a distributed but not a disentangled operator in latent space. In conclusion, our contribution is not to provide a successful learning procedure for any and all groups, but to warn against using disentangling operators as a general strategy to learn transformations, and to justify theoretically an alternative strategy to disentanglement via distributed operators in latent space to learn these transformations.
> > > > > >
> > > > > > => We will remove all ambiguous claims that may at all imply the shift operator could be used to learn all possible groups, e.g. the claim noted by the reviewer "Importantly, the shift operator does not require knowledge of the transformation in advance, only the cycle order of each group".
> > > > > >
> > > > > > => We will rewrite Section 4 (Distributed Disentanglement in Practice) as a subsection of Section 3 (Learning Transformations with Distributed Operators as an alternative to Disentanglement), and emphasize that the goal of Section 4 is only to provide empirical evidence that a distributed operator can learn affine transformations, unlike a disentangled operator which suffers from topological defects. We will emphasize in the discussion that the shift operator that we use can only model cyclic groups, or combinations of cyclic groups by stacking operators.
> > > > > >
> > > > > >  How could groups with unknown structure be learned? For the purpose of our demonstration, which is to show that distributed operators is a solution to the aforementioned  topological defects of disentanglement, we hard-code the operator in latent space to be the shift operator. However, this rigidity is not necessary and in fact a family of latent operator can be learned jointly with the encoder and decoder weights. This strategy was used successfully in Connor and Rozell for example (https://arxiv.org/abs/1912.02644), allowing them to learn a complex group with multiple subgroups corresponding to the gait of a stick figure. We don't implement this strategy in the present work because we believe that we demonstrate more clearly the advantage of distributed operators by hard-coding the operator to be the distributed and showing that unlike the disentangled operator, this operator is successfully learning the affine transformations.
> > > > > >
> > > > > > => We will add this discussion point to the paper.
> > > > > >
> > > > > >
> > > > > > _3) Recapitulating our contributions and thanking the reviewers_
> > > > > >
> > > > > > In summary, we would like to recapitulate our contributions in light of the reviewers comments and of our proposed changes: (1) We show that disentanglement via subspaces introduces topological defects for a broad family of transformations acting on images —encompassing simple affine transformations such as rotations and translations. (2) These topological defects justify the use of an alternative, more flexible approach to learning transformations via distributed operators allowing the model to be equivariant, and potentially acting on the entire latent space. (3) We theoretically and empirically demonstrate the effectiveness of distributed operators to learn simple affine transformations. Our work provides a theoretical justification for the success of a recent line of empirical work learning complex transformations via distributed operators in latent space.
> > > > > >
> > > > > > We thank the reviewers for their deep interest in our work and their sharp comments. We would like to assure them that even after the end of the discussion period, we are committed to improving the paper with any additional suggestions they might have.

---

### Official Review · AnonReviewer4 · 2020-10-27
**A very interesting paper that needs to be restructured before publication.**

**Rating:** 7
**Confidence:** 4

**Review:**

The paper first shows that existing approaches to latent space disentanglement perform poorly when the latent space topology (usually Euclidean) does not match the actual data topology, using rotation equivariance as an example. This analysis culminates in a general impossibility theorem for this type of disentanglement. The authors then propose a relaxed definition of disentanglement and show that it can be realized by means of a shift operator in latent space. Theoretical and empirical results demonstrate the superiority of the new approach. This is a very interesting idea that represents significant progress in an important problem.

Unfortunately, the current organization of the paper does not work well: the authors devote too much space (half of the paper!) to the explanation of the problem, and too little (barely one page) to its solution. This leaves the reader with many unanswered questions about how the new method works and what its crucial details are. Some of these questions are later dealt with in the appendix, but this is too late.

My main suggestion for improvement is therefore to move most of section C 3.1 to the main text and allocate the required space by shortening the motivation (up to section 3.2) and possibly the discussion of multiple transformations in section 4.3. Content that would get lost by this change should be moved to the appendix.

More minor points are:
* The authors repeatedly refer to "recent success of ... distributed operators", but do not cite and discuss any prior work. Please add appropriate references to the introduction or related work.
* Rotations and translations are continuous transformations, whereas the proposed shift operator is discrete. Does this discretization introduce rounding errors or other artifacts? How many discretization levels are needed, and how can this number be determined? Does discretization have undesirable limitations? Such potential limitations should at least be acknowledged. Ideally, these questions should be investigated experimentally (but this can be left for future work if infeasible in the present paper).
* In appendix E.1, results for rotations are an order of magnitude better than those for translations. Why is this the case?
* Figure 3E: It is hard to judge if the results align with the ground truth. Preferably, the ground truth should be displayed for reference.
* Is it necessary to design the network according to a-priory knowledge of the relevant group transformations, or can this be inferred automatically? For example, what happens if the latent space implements a group that does not correspond to any symmetry in the data?

I'm willing to raise my rating if these points (in particular, the relocation of section C 3.1) are suitably addressed in an updated version of the submission.

---

> ### Author Response · Authors · 2020-11-18
> **Response to Reviewer 4**
>
> _1) Reorganization of the paper_
>
> We thank the reviewer for his important suggestion of reorganization: the solution to the problem of disentanglement was only presented in detail in Appendix C.3.1, although it is one of the main contribution of the paper.
>
> => In the updated version of the paper, we now use the extra page allotted to us to introduce most of the material from Appendix C.3.1 in the main text.
>
>
>
> _2) References to prior work using distributed operators_
>
> The references to this prior work were not missing from the paper but were introduced in the last paragraph of the discussion: "Finally, our work lays a theoretical foundation for the recent success of a new family of methods that —instead of enforcing disentangled representations to be restricted to distinct subspaces— use operators (hard-coded or learned) acting on the entire latent space (Connor & Rozell, 2020; Connor et al., 2020; Dupont et al., 2020; Giannone et al., 2020; Quessard et al., 2020)."
>
> => To improve the visibility of these references, We add "(see discussion)" in the abstract and intro when we allude to this prior work.
>
>
> _3) The shift operator handles discrete groups only_
>
> Indeed, the shift operator we propose only handles cyclic groups with finite order (or a product of such groups). If we were to use the shift operator to model continuous transformations, a discretisation step would be needed indeed. The level of discretisation could be learned as a hyper parameter, which would increase the number of parameters to tune. This method could work if only a subset of all possible continuous transformations appears during training (which is expected using a finite training dataset), but may struggle to generalise to new values of the transformation at test-time.  It would be very interesting to investigate this in future work.
>
> The reviewer might also be interested by these references: Falorsi et al. proposes an extension of VAE with the reparametrisation trick on Lie algebra of SO(3), and Connor et al. uses the exponential map to model continuous transformations in latent space.
>
> => We thank the reviewer for pointing this out and we now acknowledge this limitation clearly in the main text.
>
>
> _4) Results are better for rotations than translations_
>
> We confirm that the MSE obtained is lower  for rotations than for translations. We believe this is due to the fact that there are more overlap between successively translated shapes than between successively rotated shapes, making learning more ambiguous and thus more difficult in the case of translations. This is also the intuition we give in the Appendix B.3 paragraph “Effect of the number of latent transformations” to explain why, in the case of translations, the weakly supervised model provides best results using a larger number of latent transformations than the ground-truth order of the group.
>
> => We have emphasized this point in the main text.
>
>
> _5) Show reconstructions and ground-truths_
>
> => We have added appendix Figures 17 and 18 showing pairs of samples and reconstructions by the stacked shift operator model in the cases of (i) translation in both x and y axes and (ii) rotations and translations in both axes. We refer to these Figures in the main text.

---

> > ### Author Response · Authors · 2020-11-18
> > **Response to Reviewer 4 (continued)**
> >
> > _6) A priori knowledge of the transformations is required_
> >
> > We believe this is partly a misunderstanding and we would like to clarify the assumptions made when using the shift operator. The shift operator is used to represent the action of the group on the latent space. Importantly:
> >
> > * While the shift operator simply computes a shift of the latent space, we use this form of operator to represent *any* finite cyclic group of affine transformations (e.g. either rotation, translation in x, translation in y). The role of the encoder is to construct a latent space where all these transformations (even rotations) can be represented as shifts. We stack shift operators to represent *any* product of such finite cyclic groups (i.e. rotations and translations combined).
> > * In the affine case that we consider, and in the supervised setting, the shift operator does not require knowledge of the transformation in advance, only the cycle order of each group (e.g. number of discrete rotation angles), which is a requirement we relax in the *weakly supervised* setting.
> > * Our study of the character of affine transformations allows us to guarantee that the shift operator respects the character of the transformation, and allows a linear equivariant model to be learned from pairs of examples for any affine transformation (see our training objectives in Eq. 9 and 10).  In addition, note that these types of distributed operators have also been shown to work empirically even in the case where character theory does not directly apply, such as or out-of-plane rotation of 3D objects (see Dupont et al.) or when the affine assumption is not made (see Connor et al.).
> >
> > => We have moved details and explanations about our shift operator from the appendix to the main text.
> >
> >
> > _7) What happens if the latent space implements a group that does not correspond to any symmetry in the data?_
> >
> > If we understand correctly, the reviewer asks what would happen if the shift operator corresponds to a transformation that is not a symmetry in the dataset. First, note that as we do not “force” the shift operator to represent a specific group, there won’t be a case where the shift operator is implemented for a specific group (e.g. scaling) but this group is in fact not in the data. We foresee two cases that might represent the issue aforementioned with a learned operator:
> >
> > * One issue that could happen is that the shift operator learns to represent a transformation (e.g. rotations) but not all objects can be rotated without the shape of the objects being modified (e.g. a glass full of water becomes empty when rotated upside down). This issue appears with most equivariant models where context or semantics are not taken into account. We refer the reviewer to this interesting recent work https://arxiv.org/abs/1911.07849 which seems to explore that question, and where the model learns to focus only on relevant transformations. We consider the integration of semantics and context as future work on equivariant models.
> > * The order of the group, decided ahead of time, might not be adapted to the transformations appearing in the data. This would indeed be problematic for the supervised shift operator. However, the weakly supervised shift operator can handle this case, by setting a large enough number of latent transformations (the hyper-parameter K_L in our paper). During learning, the weakly supervised model will use only the needed number of group elements (i.e. the order of the group) among the K_L possible group elements.

---

### Official Review · AnonReviewer3 · 2020-10-28
**Interesting idea, improper contex**

**Rating:** 3
**Confidence:** 4

**Review:**

**Summarize what the paper claims to contribute.**
The authors claims to show that disentanglement into subspaces by a continuous encoder is impossible for any finite group acting on Euclidean space
The authors claim to introduce an alternative definition of disentanglement that is more flexible and leads to a

**Strengths:**
The authors consider the problem of disentangled representation learning which is of considerable interest to the community
The authors approach the problem by imposing structure through their disentangled operators

**Weaknesses:**
The reliance of the “impossibility of disentanglement” proof seems to rely heavily on the example of the perturbed triangle. The example and its assumptions seem fairly rigid and unnatural and I am unconvinced this captures the reality of disentangled representation learning with auto-encoding networks.
The approach of adding structure by means of a transformation operator was also used in [1,2] which are cited but not compared against. Instead the authors compare against various VAEs which do not impose any external structure which does not seem particularly appropriate.
If I understand correctly, the paper seems to be based on a mischaracterization of the arguments in [3]

**Clearly state your recommendation (accept or reject) with one or two key reasons for this choice.**
Reject. See weaknesses

**Supporting arguments for your recommendation.**
While the authors tackle an interesting problem and propose an interesting solution, the arguments on which the paper is based seem flawed.

**Ask questions you would like answered by the authors to help you clarify your understanding of the paper and provide the additional evidence you need to be confident in your assessment.**
As I understand, the argument is against the utility of the *linear* disentangled representation in [3]. The more flexible definition the authors propose seems quite close to *disentangled representation* in [3], please clarify the difference.
Moreover, it seems the authors suggest the definition of disentangled representations proposed in [3] requires that subspaces corresponding to factors of variation are single dimensional (section 2) which is not the case, please clarify.
How does the approach compare against other methods, namely [1,2,4] that use structure to encourage disentangling of the representation?
Section 2 asserts that the VAE and its variants do not learn disentangled representations and uses PCA to show this is true. I expect that if this same analysis were used in the structured case, a similar result would be found, in particular, since the rotation matrix interacts with multiple dimensions of the latent code. Perhaps my intuition is incorrect, please clarify.

**Provide additional feedback with the aim to improve the paper.**
Perhaps a rewording could clarify: (Supervised Disentanglement) is composed of a 2x2 diagonal block... →  is a block diagonal matrix with a 2x2 rotation matrix in the upper left block and 1s on the remaining diagonals
(just after 11) The authors state that most deep networks are differentiable, my understanding is that the common ReLU networks are not differentiable but subdifferentiable

**Possible typos:**
(VAE, beta-VAE and CCI-VAE) the ”4s” → the ``4s”
(Dfn of a group; identity element) g_k e_G = e_G g_k = e_G → g_k e_G = e_G g_k = g_k

**Post rebuttal**
I thank the authors and other reviewers for their comments and discussion. While the direction the authors pursue is of unquestionable merit, I remain unconvinced that the work as it stands is sufficiently impactful for this venue.

[1] Falorsi, Luca, et al. "Explorations in homeomorphic variational auto-encoding." arXiv preprint arXiv:1807.04689 (2018).
[2] Connor, Marissa, and Christopher Rozell. "Representing Closed Transformation Paths in Encoded Network Latent Space." AAAI. 2020.
[3] Higgins, Irina, et al. "Towards a definition of disentangled representations." arXiv preprint arXiv:1812.02230 (2018).
[4] Cohen, Taco, and Max Welling. "Learning the irreducible representations of commutative lie groups." International Conference on Machine Learning. 2014.

---

> ### Author Response · Authors · 2020-11-18
> **Response to Reviewer 3**
>
> _1) Reliance on the example of the perturbed triangle_
>
> Our proof of the topological defects of disentanglement does not rely on this specific example, which is presented only to show the reader an example of topological defect in the specific case of rotation. In the case of rotation, any object that presents a symmetry wrt to rotation will introduce topological defects in the encoder, not just a triangle. Moreover, our proof of theorem 1 does not rely on symmetry arguments at all, but on arguments of topological isomorphisms which are much more general. Our empirical findings also confirm that disentanglement is hard in practice. Please see Appendix C.1 for the full proof of the theorem, that does not rely on the example of the perturbed triangle, nor on any symmetry argument.
>
>
> _2) Using distributed operators in latent space is not novel + comparison of our method with prior work_
>
> We do not claim that the approach to disentanglement by the means of distributed operators is novel. Our work consists in theoretically motivating *why* this approach is more suitable than traditional disentanglement via subspaces, using arguments from topology and representation theory. Our claims in the abstract and discussion are clear:
> " Our work lays a theoretical foundation for the recent success of a new generation of models using distributed operators for disentanglement."
> " our work lays a theoretical foundation for the recent success of a new family of methods that —instead of enforcing disentangled representations to be restricted to distinct subspaces— use operators (hard-coded or learned) acting on the entire latent space (Connor & Rozell, 2020; Connor et al., 2020; Dupont et al., 2020; Giannone et al., 2020; Quessard et al., 2020)."
> We do not see value in comparing our results to this prior work, as we believe that these models, similar to the one we introduce in section 4, will also be able to learn the affine transformations we learn in this paper.
>
>
> _3) Comparison to VAE is not appropriate_
>
> We do not compare the performance of our model to VAE models. We do study the failure mode of VAEs and their variants because they are a standard approach to disentanglement in the field. We agree with the reviewer that comparing VAEs with our approach would be unfair because our approach requires supervision via pairs of transformed examples and a choice of operators in latent space. However, the comparison in section 2 and table 2 is with supervised auto-encoders using operators restricted to a subspace. This comparison is fair because it is exactly the same supervised setting that we use in section 4, where we show the merits of using distributed operators in latent space.
>
>
> _4) If I understand correctly, the paper seems to be based on a mischaracterisation of the arguments in [3]_
>
> It is not at all clear to us how our work would be mischaracterizing Higgins et al. arguments. To clarify, our work builds on the definition of disentanglement of Higgins et al. but we extend their work in several ways. First, we show that traditional disentanglement introduces topological defects (i.e. discontinuities in the encoder), even in the case of simple affine transformations. Second, we conceptually reframe disentanglement, allowing equivariant operators to act on the entire latent space, so as to resolve these topological defects. Finally, we show that models equipped with such operators successfully learn to disentangle simple affine transformations.
>
>
> _5) PCA analysis on our approach would present a flat eigenspectrum_
>
> Indeed, a PCA analysis of the latent space under our distributed approach to disentanglement would present a flat eigenspectrum. But in our approach, contrarily to the case of VAE where the PCA analysis is applied, we learn to map the transformation present in the data to a known operator acting on the latent space, and so we know how to emulate the transformation learned in latent space, despite the fact that it is distributed. This is not the case for VAEs, where there is no known operator in latent space that is equivariant to the transformation learned. The PCA analysis is thus relevant in the VAE approach, but not in our approach to disentanglement via distributed operators.
>
> _6) ReLUs are not differentiable_
>
> The reviewer is correct to point out that RELU networks are not differentiable everywhere, and therefore, as we note in the main text, our proof about the impossibility of obtaining an invariant subspace to a transformation on Euclidean space does not hold for RELU networks. However, our more general Theorem 1 proven in Appendix C.1 does not rely on differentiability but on the continuity of the encoder and thus holds even in the case of RELU networks.
>
>
> _7) Additional Feedback_
>
> We thank the reviewer for the additional feedback that we will use to clarify some aspects of the paper, and for finding a typo in Appendix A.

---

### Official Review · AnonReviewer1 · 2020-11-03
**Review of "Addressing the Topological Defects of Disentanglement"**

**Rating:** 6
**Confidence:** 3

**Review:**

Summary: The authors proposed a new way to disentangle affine transformations without topological defects. This paper made several theoretical contributions including a new definition of disentanglement and demonstration of the topological defects in existing disentanglement methods. Experimentally, this paper showed how their proposed shift operator model is powerful when dealing with topological defects.

Disentanglement is a relatively challenging task due to the lack of clear definition and the lack of a robust evaluation method. The authors did a good job providing new theoretical definitions and providing empirical and qualitative results to support their claims. The main weakness of the paper is the lack of quantitative metrics to evaluate their approach and compare with others. In addition, the model doesn’t appear to be very flexible as it requires that the transformation is known in advance.

Strengths:
+ Overall, the paper is well written and contains a good review of advances in the theory of disentanglement.
+ The idea of addressing topological defects for disentanglement appears novel.
+ Using operators on the entire latent space is a new direction for the study of disentanglement. The authors’ viewpoint that “isolating factors of variation” is different from “mapping these factors into distinct subspaces”, and how they propose a new definition based on this viewpoint is interesting.


Weaknesses:
- Lack of quantitative evaluation metrics. The MSE in the appendix is not enough for quantifying disentanglement.
- Since this paper focuses on disentanglement, at least Factor-VAE, one of the other representative disentanglement VAE models should be considered when doing the model evaluation.
-  Baseline models should be optimized in a more comprehensive manner (e.g., currently the selection of beta is {4, 10, 100, 1000} and latent dimension is {10, 30}). It’s unclear whether these models have been well optimized, or what measures are used to optimize the models for this task.
- Because the method requires that the transformation is known in advance, this limits the flexibility of the approach.
- How different transformations impact each other is not shown experimentally - there is only an example on Fig 3E showing some visual results, but this should be elaborated on further given the goal of the paper.

Minor points:
- The complex version of the shift operator is used. It would be interesting to show another version and their differences.
- Latent traversals results appear to be rather sparse. It would be interesting to show how the variation exists inside the model via dense traversals and the computing of generated images variation with different latent traversals.
- Rotations may be more challenging to learn. 2000 examples may be insufficient for the model to learn this transformation correctly.

---

> ### Author Response · Authors · 2020-11-18
> **Response to Reviewer 1**
>
> _1) Lack of quantitative metrics of disentanglement_
>
> We agree with the reviewer that it is interesting to quantify disentanglement beyond the MSE of the transformed reconstructions. However, traditional metrics of disentanglement (such as mutual information gap) cannot be applied with distributed operators in latent space. Indeed, traditional disentanglement metrics are not appropriate as they describe how well factors of variation are restricted to subspaces---in contrast to our proposed framework using distributed latent operators. To further quantify the evaluation of the shift operator, we thus compute LSBD, a measure of disentanglement appropriate for distributed operators proposed by the recent ICLR submission https://openreview.net/forum?id=YZ-NHPj6c6O. LSBD measures how well latent operators capture each factor variation, allowing us to quantify disentanglement in the setting of distributed operators.
>
> => Using this new metric, we further quantify the advantage of the shift operator with LSBD of 0.0020 versus the disentangled operator with LSBD of 0.0106 for the models in Figure 3.A and 3.B. We now include these results in Section 4.1 and Appendix E.1 of the manuscript. The LSBD measure confirms our existing qualitative results (Figure 3 and Appendix E.2) and quantitative MSE measures in Appendix E.1.
>
>
> _2) Comparison with FactorVAE_
>
> We already implemented three models commonly used for disentanglement: VAE, Beta-VAE, and CCI VAE. However, we agree with the reviewer that it would also be interesting to implement FactorVAE as an additional baseline model.
>
> => We are in the process of implementing FactorVAE as an additional baseline model. We commit to add the FactorVAE baseline to the final version of the paper.
>
>
> _3) Hyper-parameter optimization for baseline models_
>
> We agree that optimizing hyper-parameters is important for baseline models. Our hyper-parameter optimization sweeps over both sets of model hyper-parameters used to successfully disentangle factors of variation in the CCI-VAE paper by Burgess et al. Additionally, we sweep over all combinations of model hyper-parameters (beta, latent dimension) and training parameters (learning rates, batch sizes, and random seeds), amounting to a sweep of >1000 models per baseline (see Appendix B.4). We select the model minimizing reconstruction MSE on a held-out validation set separate from the test and training sets.
>
> _4) A priori knowledge of the transformations is required_
>
> We believe this is partly a misunderstanding and we would like to clarify the assumptions made when using the shift operator. The shift operator is used to represent the action of the group on the latent space. Importantly:
>
> * While the shift operator simply computes a shift of the latent space, we use this form of operator to represent *any* finite cyclic group of affine transformations (e.g. either rotation, translation in x, translation in y). *The role of the encoder is to construct a latent space where all these transformations (even rotations) can be represented as shifts. *We stack shift operators to represent *any* product of such finite cyclic groups (i.e. rotations and translations combined).
> * In the affine case that we consider, and in the supervised setting, the shift operator does not require knowledge of the transformation in advance, only the cycle order of each group (e.g. number of discrete rotation angles), which is a requirement we relax in the *weakly supervised* setting.
> * Our study of the character of affine transformations allow us to guarantee that the shift operator respects the character of the transformation, and allows a linear equivariant model to be learned from pairs of examples for any affine transformation (see our training objectives in Eq. 9 and 10).  In addition, note that these types of distributed operators have also been shown to work empirically even in the case where character theory does not directly apply, such as or out-of-plane rotation of 3D objects (see Dupont et al.) or when the affine assumption is not made (see Connor et al.).
>
> => We have moved details and explanations about our shift operator from the appendix to the main text.

---

> > ### Author Response · Authors · 2020-11-18
> > **Response to Reviewer 1 (continued)**
> >
> > _5) How different transformations impact each other_
> >
> > Regarding how different transformations impact each other, we provide experimental results in Figure 3E that shows exemplar results of our model for a discrete version of the Special Euclidean Group on rotated-translated shapes. Figure 3D shows the case of translation in both x and y axes at the same time. Additionally, appendix Figure 14b shows results of this model on rotated-translated MNIST and Figure 16 shows results on rotated-translated shapes when the semi-direct structure of the group product is not respected. In Table 2, we also report MSE for the stacked shift operator on both datasets.
> >
> >
> > In addition to these results showing how we can successfully learn a combination of transformations, the paragraph “Insight from representation theory on the structure of hidden layers” in the main text describes the theoretical challenges of dealing with rotations and translations happening together. We note that since rotations and translations do not commute, the correct operator cannot be diagonal, otherwise two operators corresponding to two elements would commute. Indeed, as we show in Appendix D.4 the resulting operator for the discrete finite Special Euclidean case has a block matrix form based on representations of both translations and rotations. We do not directly use this form in our model, but instead use the stacked version of the shift operator, with intermediate linear layers in-between diagonal shift operators. However, Appendix D.4 gives us insight about the form that intermediate layers should take after training: we prove that the intermediate layers of our stacked model should have a block-diagonal form.
> >
> > => We have changed the main text to emphasise these results. We have also added appendix Figures 17 and 18 showing pairs of samples and reconstructions by the stacked shift operator model in the cases of (i) translation in both x and y axes and (ii) rotations and translations in both axes.
> >
> >
> > _6) Complex and real versions of the shift operator_
> >
> > We also use the real version of the shift operator in Figure 3A, as described in the main text. In all our experiments, we see no difference in the results between the real and complex operators, as predicted by the theory.
> >
> > _7) Dense latent traversals_
> >
> > We apply the same latent traversal procedure used in popular disentanglement methods (Beta-VAE and CCI-VAE). We further extend the traversal range from [-3, 3] to [-6, 6] to ensure that we capture the full space of possible variation.
> > =>We also now include 3 additional figures in Appendix E.2, using a more dense latent traversal with 50 plots per latent dimension.
> >
> >
> > _8)Rotations are challenging to learn_
> >
> > We do not entirely rule out the possibility that with a bigger network and many more samples, we could learn rotation with the traditional disentanglement approaches, as acknowledged in the main text. What we do show, however, is that the function learned must be highly discontinuous with arguments from topology. We also propose a new approach to disentanglement which is successful at learning the rotation transformations with as few as 2000 samples, by respecting the topology of the transformation to disentangle.

---

### Official Review · AnonReviewer5 · 2020-11-07
**Interesting conceptual formulation, not practically developed**

**Rating:** 6
**Confidence:** 4

**Review:**

This paper presents the idea that the current formulation of disentangled latent representations of data that have been presented are implausible in the sense that the factors are often not actually independent and cannot be learned or generated as independent. Instead the authors put forth the idea of transformations of data that are equivariant to the latent space representation as a formulation of disentangled factors. The authors use group theoretical constructs such as shift and rotation operators to show that a latent space representation should be equivariant such transformations. In other words, if a latent space representation is rotated, it should still reconstruct correctly, because the reconstruction loss should be trained on a rotated version of the image.

The key strengths of this paper are the examples that showcase the lack of ability to learn independent latent factors. Figure 1 displays the failure to learn rotation as a factor in the MNIST digit dataset. Figure 2 is even more convincing in that it shows that the orbits of the different factors cannot be mapped to one another and thus cannot be truly independent.

Second, I believe that the idea that is better stated in the introduction  on how disentanglement can be framed is valuable: “ In this framework, the factors of variation are different subgroups  acting on the dataset, and the goal is to learn representations where separated (of the data) subspaces are equivariant  to distinct subgroups.” Theoretically the authors are proposing an operational view of the latent factors as separate transformations on the data, and the representation as having subspaces equivariant to the transformations. Definition 1 is trying to state the same idea but is much less clear to the average ML reader

More generally, the authors should work harder to communicate this to the ML audience. The group theoretical background from the appendix should be in the background section, particularly the idea of equivariance and group operations.

The key weakness is that their new formulation of disentanglement is that it is definitional does not give a plan of how this should be done. Based on their description it seems as if the dataset has to come with a set of known operations on the data (like rotations) that are equivariant. How would such operations be learned de novo from the data? It seems as if the framework requires learning two things separately 1. A latent representation of the data 2. A set of equivariant operations on the data (that are perhaps cyclic generators of an orbit). It is not clear how this would be learned.

---

> ### Author Response · Authors · 2020-11-18
> **Response to Reviewer 5**
>
> _1) Definition 1 is unclear_
>
> => We updated our definition of disentanglement in the main text to reflect the comments of the reviewers:
>
> “Definition 1. A representation is disentangled with respect to a set of transformations, if there is a family of controllable operators, potentially acting on the entire representation, where each operator corresponds to the action of a single transformation and the resulting model is equivariant.
>
> These operators are controllable in the sense that they have an explicit form, thus allowing the user to manipulate the latent representation by applying the operator. This definition, more flexible than traditional disentanglement in the choice of the latent operators, obeys to the same desiderata of identification and isolation of the factors of variations present in the data.“
>
>
> _2) Reorganisation of the paper by adding background into main_
>
> We thank the reviewer for pointing out the opportunity for additional background. We made several changes to ensure both the idea of equivariance and the use of group theory is clearly introduced for a general ML audience.
>
> =>We reformulated Definition 1 without reliance on group theory, added a paragraph introducing equivariance (formally and informally in Section 3.3), and included additional background on the use of group theory for describing transformations in Section 3.4.
>
>
> _3) A priori knowledge of the transformations is required_
>
> We believe there is partly a misunderstanding and we would like to clarify the assumptions made when using the shift operator. The shift operator is used to represent the action of the group on the latent space. Importantly:
>
> * While the shift operator simply computes a shift of the latent space, we use this form of operator to represent *any* finite cyclic group of affine transformations (e.g. either rotation, translation in x, translation in y). *The role of the encoder is to construct a latent space where all these transformations (even rotations) can be represented as shifts. *We stack shift operators to represent *any* product of such finite cyclic groups (i.e. rotations and translations combined).
> * In the affine case that we consider, and in the supervised setting, the shift operator does not require knowledge of the transformation in advance, only the cycle order of each group (e.g. number of discrete rotation angles), which is a requirement we relax in the *weakly supervised* setting.
> * Our study of the character of affine transformations allow us to guarantee that the shift operator respects the character of the transformation, and allows a linear equivariant model to be learned from pairs of examples for any affine transformation (see our training objectives in Eq. 9 and 10).  In addition, note that these types of distributed operators have also been shown to work empirically even in the case where character theory does not directly apply, such as or out-of-plane rotation of 3D objects (see Dupont et al.) or when the affine assumption is not made (see Connor et al.).
>
> => We have moved details and explanations about our shift operator from the appendix to the main text.

---

### Author Response · Authors · 2020-11-18
**General response to the Reviewers**

We thank the reviewers for their time and thoughtful reviews. Their insightful comments helped improve the quality of the paper. All reviewers recognize the importance of the topic we address. Furthermore, reviewers noted the merits of our work by acknowledging the relevance of the topological flaws of disentanglement we uncovered (R1, R4, R2 and R5), the pertinence of our proposed relaxed definition of disentanglement (R1, R4 and R5), and the effectiveness of our shift operator solution for the disentanglement of affine transformations (R1, R2, R4, and R5).

The reviewers had common clarification questions and suggestions to improve the paper. Based on the reviewers’ comments, we incorporated many suggestions and clarified our contribution in several ways.

First, we clarified the assumptions of our proposed shift operator. Most important, we would like to clarify here (and have in the text) that our proposed shift operator *does not* require knowledge of the transformation to be learned in advance (R1, R4 and R5). In the case we consider (affine discrete and cyclic transformations), the only knowledge needed in the supervised setting is the cycle order of each group (e.g. number of discrete rotation angles), which is a requirement we relax in the weakly supervised setting. To clarify this point, we have moved details and explanations about our shift operator from the appendix to the main text, and added some clarifications to the main text.

Second, reviewers had excellent suggestions concerning the organization of the paper (R4 and R5), which we have also taken into account and addressed in the resubmission. Third, we improved the clarity of our definition of disentanglement (R2 and R5).

Finally, we further describe the connections and differences between our work and existing methods (R1, R2, R3).

We replied to each reviewer’s questions and comments individually. We uploaded a revised version of our paper, and in our replies to each reviewer we indicate our changes to the paper with the “=>” symbol. We are eager to discuss any further concern that the reviewers might have and thank them in advance for their time and consideration.

---

### Decision · Program_Chairs · 2021-01-07
**Final Decision**

**Decision:**

Reject

**Comment:**

This is a borderline case (quite comparable to the other borderline case in my batch). The paper has received careful reviews and based on my weighting of the different arguments I arrive at an average score between 5.75 and 6.. The authors present some worthwhile ideas related to disentanglement that deserves more attention and that could spark more research in this direction. At the same time, the level of novelty and significance of this work remains a bit limited. Taken together the paper is likely not compelling enough to be among the top papers to be selected for publication at ICLR.